# Selective therapeutic strategy for p53-deficient cancer by targeting dysregulation in DNA repair

Justin Zonneville [1], Moyi Wang[1], Mohammed M. Alruwaili [1,2], Brandon Smith [1], Megan Melnick [1], Kevin H. Eng [1], Thomas Melendy [3], Ben Ho Park[4], Renuka Iyer[5], Christos Fountzilas [5] & Andrei V. Bakin [1✉]

Breast carcinomas commonly carry mutations in the tumor suppressor p53, although therapeutic efforts to target mutant p53 have previously been unfruitful. Here we report a selective combination therapy strategy for treatment of p53 mutant cancers. Genomic data revealed that p53 mutant cancers exhibit high replication activity and express high levels of the Base-Excision Repair (BER) pathway, whereas experimental testing showed substantial dysregulation in BER. This defect rendered accumulation of DNA damage in p53 mutant cells upon treatment with deoxyuridine analogues. Notably, inhibition of poly (ADP-ribose) polymerase (PARP) greatly enhanced this response, whereas normal cells responded with activation of the p53-p21 axis and cell cycle arrest. Inactivation of either p53 or p21/*CDKN1A* conferred the p53 mutant phenotype. Preclinical animal studies demonstrated a greater antineoplastic efficacy of the drug combination (deoxyuridine analogue and PARP inhibitor) than either drug alone. This work illustrates a selective combination therapy strategy for p53 mutant cancers that will improve survival rates and outcomes for thousands of breast cancer patients.

[1] Department of Cancer Genetics and Genomics, Roswell Park Comprehensive Cancer Center, Buffalo, NY, USA. [2] Medical Laboratory Technology Department, Northern Border University, Arar City, Saudi Arabia. [3] Department of Microbiology & Immunology and Biochemistry, University at Buffalo, Buffalo, NY, USA. [4] The Breast Cancer Research Program, Vanderbilt-Ingram Cancer Center, Vanderbilt University Medical Center, Nashville, TN, USA. [5] Department of Medicine, Roswell Park Comprehensive Cancer Center, Buffalo, NY, USA. ✉email: andrei.bakin@roswellpark.org

Over three million women are living with breast cancer that takes over 40,000 lives each year, making it the second deadly cancer among women in the United States[1,2]. Molecular targeted therapies are available for the treatment of breast cancers that are positive for estrogen and progesterone receptors (ER/PR), and human epithelial growth factor receptor HER2[3]. Nearly 20% of breast cancers are negative for ER/PR/HER2 markers and constitute a group of triple-negative breast cancers (TNBCs) that lacking targeted therapy and routinely are treated with chemotherapy regimens[3,4]. Despite the high initial response, TNBCs frequently relapse and progress to metastatic disease with worse overall survival[5,6], emphasizing a need for better therapeutic options.

TNBCs largely overlap (>80%) with basal-like breast cancers defined by molecular profiling[7]. TNBCs commonly carry somatic gene alterations in the tumor suppressor p53 pathway (*TP53*, about 80−85%; *MDM2/MDM4*, amplification, about 6%), while other important mutated genes include tumor suppressors *RB1* and *PTEN*[8]. A distinct group of TNBC (<5%) includes carriers of somatic or germline mutations in *BRCA1/BRCA2* genes, which function in homologous recombination (HR)[9]. *BRCA1/2*-mutant tumors are intrinsically sensitive to inhibitors of poly (ADP-ribose) polymerase (PARP)[10,11], and this notion was translated to the clinic for treatment of *BRCA*-deficient patients[12,13]. The majority of TNBCs, though, carry wild-type *BRCA*1/2 and do not respond to PARP inhibitor[14].

The tumor status of *TP53* is largely ignored in the clinical management of patients with breast cancers, although decades of research clearly implicate p53 in the response to DNA damage through multiple mechanisms including a direct interaction with DNA repair machinery[15]. Despite of immense information on the functional consequences of *p53* mutations, therapeutic efforts targeted to mutant p53 have been largely unfruitful[16,17]. Notably, a synthetic lethal effect associated with the G2 checkpoint vulnerability of p53 mutant tumors was explored with Chk1, WEE1, and PLK1 inhibitors[16]. Nonetheless, there is no Food and Drug Administration (FDA) approved drug with promising clinical activity against p53 mutant tumors at present.

In this study, we investigated genetic-based vulnerabilities in breast carcinomas to identify targets for therapeutic intervention. We discovered substantial dysregulation in base excision repair (BER) in p53 mutant cancer cells that lead to accumulation of DNA damage upon treatment with nucleotide analogues. Based on this finding, we developed a combination therapeutic regimen that selectively targets p53-mutant breast cancer. In preclinical models, the combination of FDA-approved nucleotide analogue with a PARP inhibitor (PARPi) showed greater efficacy in inhibition of tumor growth and metastases than either drug alone. This study illustrates a selective synthetic lethality strategy for the treatment of breast cancer by means of exploiting DNA repair dysfunction of p53 mutant cancer cells.

## Results

### Activation of DNA repair pathways in TNBC.
Clinical behavior of breast cancers is linked to high proliferative activity[18] and mutational burden[19,20]. We explored the expression of replication-related genes (RRGs) in breast cancer (BC) subtypes using The Cancer Genome Atlas (TCGA) data[21]. Genomic data showed that TNBC/Basal-like cancers (TNBC thereafter) exhibit high expression of RRGs (S- and M-phase cell cycle; *t*-test *P* < 0.001), indicating elevated replication activity (Fig. 1a, b and Suppl. Fig. 1a). DNA replication consumes substantial energy and nucleotide resources[22,23], and may imbalance the pool of nucleotides, causing replication stress and increasing mutational burden[22,24–26]. Consistent with this notion, TNBCs showed high

mutational burden, while only luminal A cases (low RRG expression) had a distinctly low tumor mutation burden (Fig. 1c, *t*-test *P* < 0.001). Metastatic TNBCs also showed higher mutational burden compared to other BC subtypes[20].

TNBCs exhibit a prevalence of C-to-T transition[8], typically associated with misincorporation or modification of nucleotides[27]. Misincorporated or modified nucleotides are normally removed by DNA repair mechanisms such as mismatch repair (MMR) and base excision repair (BER), while inadequate activity or genetic alterations in these mechanisms may increase the mutational burden[27–29]. To address this idea, we assessed genomic data for BER and MMR genes. Unsupervised clustering revealed that TNBCs express both BER and MMR genes at the highest level compared to other subtypes (Suppl. Fig. 1b), with a high correlation between MMR and BER gene sets ($\rho = 0.88$, Fig. 1d). These findings indicated that TNBCs have high replicative activity along with increased expression of BER and MMR pathways.

Notably, expression of BER and MMR genes was highly elevated in p53-mutant breast cancers (Fig. 1e and Suppl. Fig. 1c), while genetic alterations in these repair genes were infrequent events (Fig. 1e). DNA glycosylases involved in the repair of uracil and base-modified nucleotides, i.e., UNG, TDG, and MUTYH, were highly expressed in TNBC (Fig. 1e). These data suggested that p53 mutant (p53mt) cancers may experience dysregulation or impediment in BER and MMR mechanisms even in the absence of genetic alterations in DNA repair genes. This idea is supported by the high mutational burden in p53mt breast cancers (Suppl. Fig. 1e) and the critical role of p53 in DNA repair[15].

### BER-mediated DNA repair in p53 mutant breast cancer cells.
Tumors with compromised p53 function are characterized by increased levels of C-to-T transitions[30]. Mismatched nucleotides and indels both arise during DNA replication and can be removed by MMR mechanisms. The mutation signature of MMR-deficient cancers is characterized by microsatellite instability, a result of indel repair deficiency[31]. Conversely, the BER system removes DNA base lesions leading to C-to-T transition such as deamination (uracil), oxidation (e.g., 8-oxoguanine), or alkylation (e.g., 3-methyladenine and 6-ethyl-guanine)[32]. Based on this rationale, we examined BER repair capacity in a breast cancer MDA-MB-231 cell line carrying mutant p53 (R280K) and a normal breast epithelial MCF10A cell line with wild-type (wt) p53. Neither cell lines carry genetic alterations in the BER or MMR pathways, based on genomic data; while MDA-MB-231 cells do express high levels of UNG and RRGs compared to MCF10A cells (Suppl. Fig. 2).

The BER-mediated repair was assessed by measuring genomic 5-ethynyl-2'-deoxyuracil (EdU) at various washout timepoints following a pulse of 5-ethynyl-2'-deoxyuridine (EdUrd) (Fig. 2). Genomic EdU was labeled with fluorophore using click-it chemistry and then scored by flow cytometry or microscopy. Both p53wt and p53mt cell lines incorporated comparable levels of EdU and incubation with hydroxyurea, a ribonucleotide reductase inhibitor, blocked EdU incorporation into DNA (Suppl. Fig. 3a). Flow cytometry revealed that the EdU⁺ fraction was reduced at 24 h in MCF10A cells, whereas MDA-MB-231 cells retained the EdU⁺ fraction for a prolonged time (Fig. 2a, b). This finding was confirmed by fluorescence microscopy (Suppl. Fig. 3b, c). Thus, both methods indicated that non-tumor p53wt cells efficiently removed uracil analogue from DNA, while this activity was diminished in p53mt cancer cells.

We noted that the EdU⁺ fraction in the p53mt cell line was increased by ~25% (*P* < 0.05) at 24 h after EdUrd-pulse (Fig. 2a, b), while DNA replication was not paused as it was observed in p53wt MCF10A cells, based on the cell cycle data (Fig. 2c). The increase in genomic EdU in p53mt cells might be

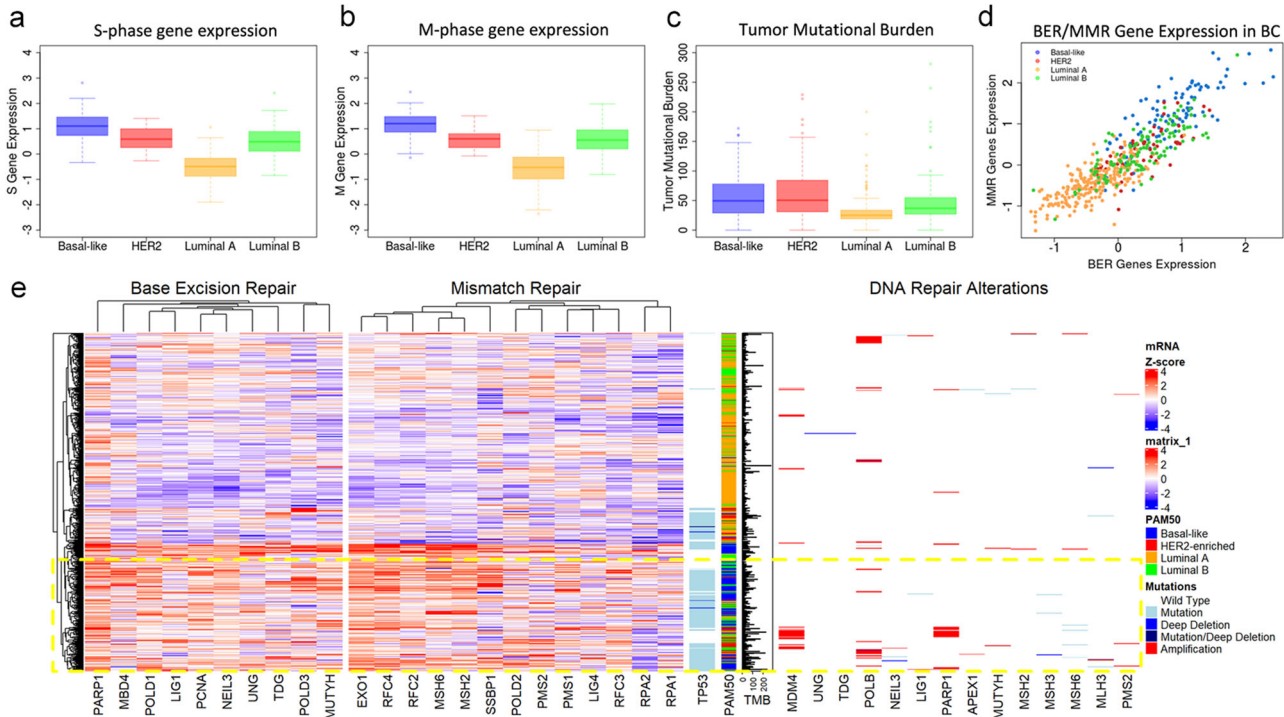

**Fig. 1 Activation of DNA replication and DNA repair pathways in TNBCs. a, b** Expression of replication-related genes (S-phase and M-phase) in breast cancer subtypes (TCGA BC dataset). Gene lists are designed using the Cyclebase_3.0 database. **c** Tumor mutational burden in breast cancer subtypes (TCGA BC dataset). **d** MMR and BER pathway scores stratified by PAM50 classification. Gene lists are derived from KEGG. **e** Expression of individual genes in BER and MMR pathways is elevated in p53-mutant tumors in predominantly TNBC/Basal-like cancer cases (highlighted by the yellow box).

caused by cycles of futile BER-mediated repair, a well-known phenomenon for fluorinated uridine analogues[33]. Monophosphate metabolites of both EdUrd and fluorinated analogues (EdUMP and FdUMP) inhibit thymidylate synthase[34,35], thereby stimulating incorporation of uridine analogues into DNA and, subsequently, activating BER repair[33,36]. Multiple cycles of BER repair lead to the accumulation of apyrimidinic sites (Fig. 2e), disruption of A-T pairs, and DNA breaks[33]. This notion was validated using Hoechst dye that selectively binds to A-T-rich regions[37]. Flow cytometry revealed expansion of the EdU+ fraction with low Hoechst fluorescence in p53mt MDA-MB-231 cells at 24 h (Fig. 2a–d), indicating a reduction in Hoechst binding to DNA in agreement with disruption of A-T pairs. Next, we confirmed these findings in TNBC cell lines with mutant p53 (R273H) MDA-MB-468 and wild-type p53 CAL51 (Suppl. Fig. 4). Mutant p53 cells showed a substantial delay in genomic EdU removal and a shift in Hoechst fluorescence indicating A-T pair disruption compared to p53wt CAL51 cells (Suppl. Fig. 4c, d).

DNA damage was evaluated by examining phosphorylation of H2AX at Ser139 (γH2AX) that marks DNA double-strand breaks (DSBs)[38]. In p53wt MCF10A cells, EdUrd-pulse caused a transient increase of γH2AX levels, whereas p53mt MDA-MB-231 cells accumulated DNA damage at 48 h (Fig. 2f). In p53wt cells, EdUrd-pulse activated p53 (phospho-Ser15) and increased the levels of cell-cycle inhibitor p21, a transcriptional target of p53, while p21 was not induced in p53mt cells. Consistent with this finding, EdUrd-pulse interrupted replication in p53wt MCF10A cells, while p53mt MDA-MB-231 cells persisted in the S phase (Fig. 2c). Notably, EdUrd-pulse caused a transient formation of the TS inhibitory complex in MDA-MB-231 cells, indicated by a shift in TS mobility, and this was declined at 6 h thereafter (Fig. 2f). The TS inhibition was temporally separated from the DNA damage signal (γH2AX) and p53 activation,

suggesting that activation of p53 is mediated by DNA damage signaling.

Taken together, these data indicated that p53wt cells effectively remove genomic ethynyl-uracil through the BER mechanism. Concurrently, EdUrd-pulse induced the p53−p21 axis and G1 arrest in p53wt cells. In contrast, p53mt cancer cells did not stop DNA replication in response to EdUrd, and this resulted in accumulation of genomic uracil analogue and increased DNA damage, based on the flow cytometry and γH2AX data (Fig. 2a, f). The Hoechst flow cytometry data suggested that this response is associated with futile DNA repair[33] and persistence of DNA replication due to inactive the p53−p21 axis.

**The p53−p21 axis limits DNA damage induced by the deoxyuridine analogue.** Next, we tested whether continuous treatment with EdUrd or clinical deoxyuridine analogues causes DNA damage selectively in p53mt cells. EdUrd induced the p53−p21 axis and γH2AX level at 24 h in MCF10A cells, and the γH2AX level declined at 48 h, while p53mt cancer cells responded to EdUrd with continued accumulation of DNA damage signal (Fig. 3a). EdUrd induced phosphorylation of p53 in both cell lines, while the TS-inhibitory complex was induced only in p53mt MDA-MB-231 cells (Fig. 3a). Floxuridine (5-fluoro-2'-deoxyuridine, FdUrd), a clinically relevant deoxyuridine analogue, induced a transitory DNA damage and activation of the p53−p21 axis in p53wt cells, while p53mt cancer cells accumulated DNA damage signal as it was observed for EdUrd (Fig. 3b). These findings were verified in mouse mammary carcinoma p53wt (EMT6) and p53-null (4T1) cell lines (Fig. 3c), both are established TNBC models. Further tests in p53wt cancer cell lines, breast cancer CAL51 (TNBC), and lung cancer A549, demonstrated activation of the p53−p21 axis and a transient induction of DNA damage (Fig. 3d, e). Conversely, breast cancer (TNBC)

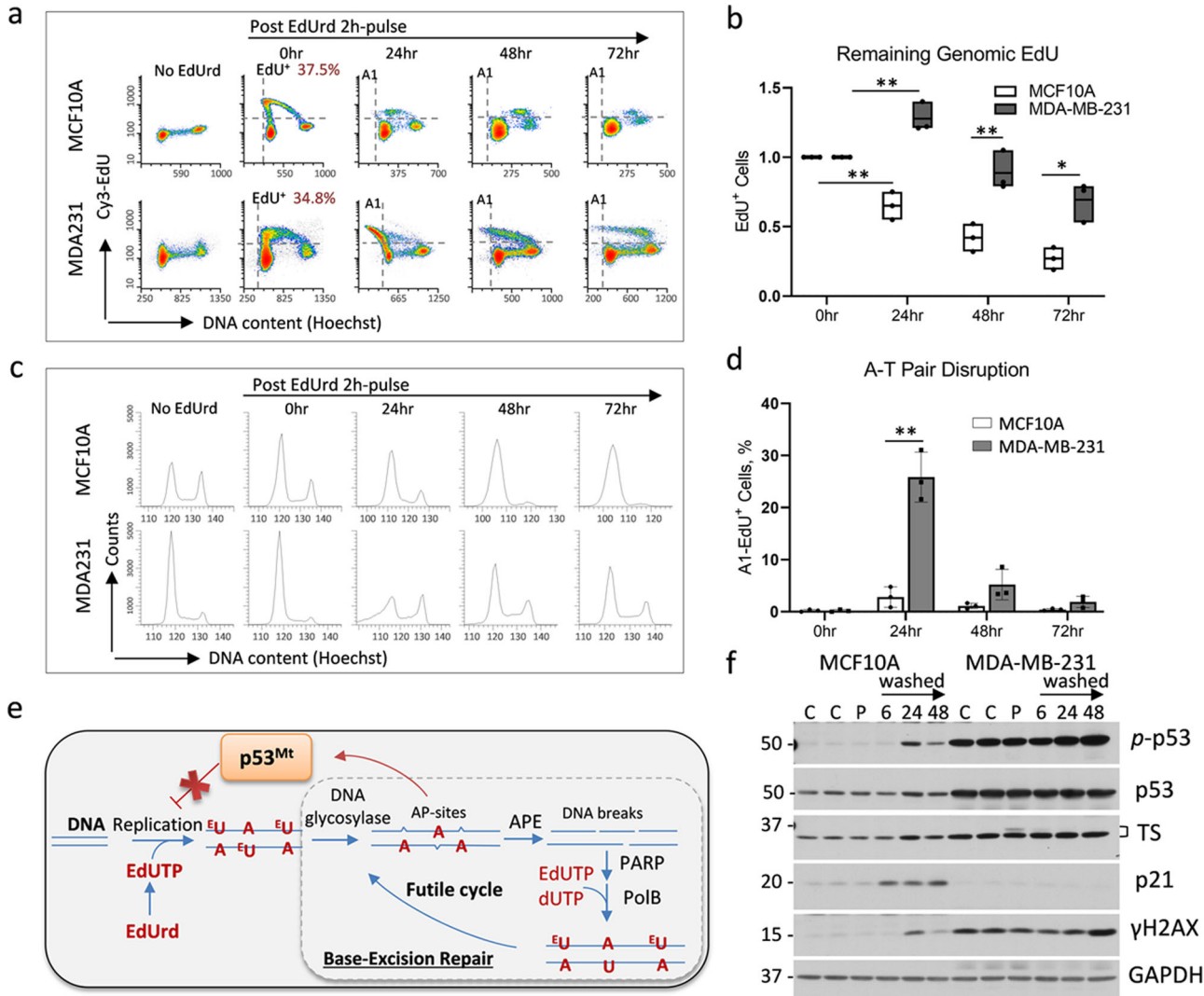

**Fig. 2 Removal of genomic ethynyl-deoxyuracil (EdU) by BER in non-tumor and tumor cells. a, b** Breast epithelial p53wt MCF10A and breast cancer p53mt MDA-MB-231 cell lines were pulse-labeled with 5-ethynyl-2′-deoxyuridine (EdUrd) for 2 h followed by wash and incubation for the indicated time. Cells were fixed and EdU was stained using click-it chemistry and analyzed by flow cytometry with Hoechst for DNA content. EdU-positive populations were scored relative to initial levels. **c** Cell cycle data for cells treated as in (**a**). **d** EdU⁺ cells with reduced Hoechst fluorescence (upper-left quartile). The comparison was made using the Log-rank test (*, $P < 0.05$; **, $P < 0.01$). **e** Scheme illustrates the model of EdUrd-induced futile cycle of BER-mediated DNA repair and the role of p53. **f** Immunoblot data in MCF10A and MDA-MB-231 cells after 2 h EdUrd-pulse (P), and 24−48 h after wash and incubation in EdUrd-free media.

MDA-MB-468 cell line carrying mutant p53-R273H accumulated DNA damage in response to FdUrd (Fig. 3d).

Then, we tested whether inactivation of the p53−p21 axis in p53wt cells will lead to the accumulation of DNA damage in response FdUrd. p53 contributes to multiple aspects of DNA repair, including BER[15,39], while the CDK inhibitor p21 mediates G1 and intra-S arrest in response to DNA damage by binding to the proliferating cell nuclear antigen (PCNA) and inhibiting DNA replication[40]. In addition, p21 may regulate cell cycle and DNA repair pathways[41,42]. As expected, depletion of p53 by siRNA ablated induction of the p53−p21 axis by FdUrd in MCF10A cells (Fig. 3f). Importantly, p53-depleted cells responded to FdUrd treatment with the accumulation of DNA damage (γH2AX), whereas scramble-control cells exhibited a temporal increase in γH2AX and induction of p21 (Fig. 3f).

Next, the role of p21 was examined using p21 knock out (ko) MCF10A cells[43]. Treatment with FdUrd induced phosphorylation and total p53 levels in control and p21ko cells,

while, as expected, p21 was induced only in control cells (Suppl. Fig. 5). FdUrd induced a transient DNA damage in control p21wt cells, whereas p21ko cells accumulated DNA damage. We confirmed the ability of p53 to regulate transcriptional targets such as MDM2 in p21ko cells by using Nutlin-3A, which disrupts p53−MDM2 interactions[44]. Nutlin-3A activated p53 and increased MDM2 levels in both p21wt and p21ko cells but did not induce DNA damage (Suppl. Fig. 5). Note, Nutlin-3A did not increase DNA damage in both p53wt and mutant cells (Fig. 3d).

The formation of DNA breaks in response to FdUrd was confirmed by microscopic evaluation of γH2AX and RAD51 foci (Fig. 3g, h) that are formed around DNA breaks[38]. Quantification of γH2AX foci fluorescence showed a transient increase in the fluorescence intensity at 24 h in p53wt MCF10A cells, whereas the fluorescence was steadily accumulated in p53mt MDA-MB-231 cells (Fig. 3h). Likewise, FdUrd strongly induced the fraction of cells with RAD51 foci in p53mt cancer cells, whereas in p53wt

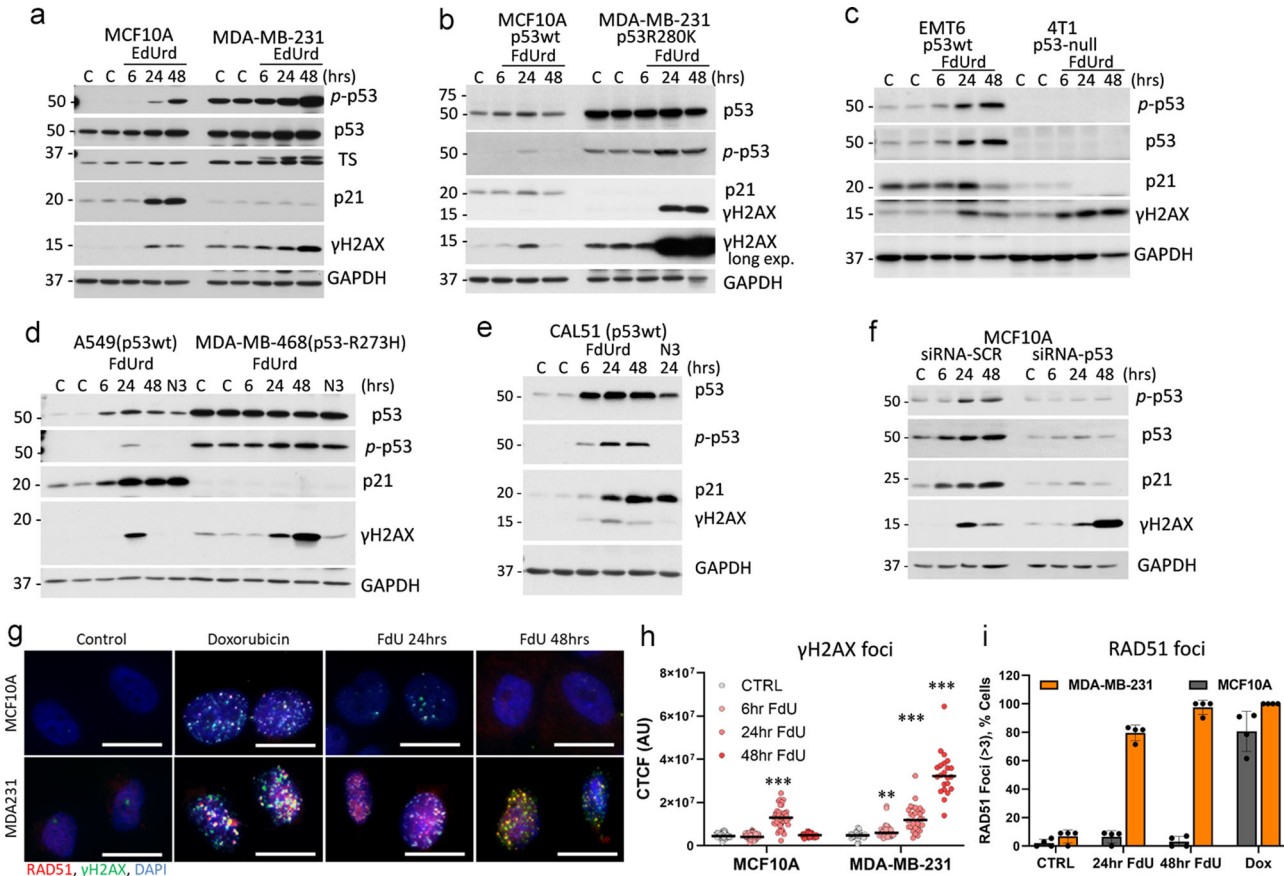

**Fig. 3 p53−p21 signaling controls DNA damage in response to deoxyuridine analogues.** Immunoblot analysis of the response to ethynyl-deoxyuridine (EdUrd, 5 μM) (**a**) or to floxuridine (5-fluoro-2'-deoxyuridine, FdUrd, 5 μM) (**b**−**e**) in p53wt MCF10A, CAL51, EMT6 and p53mt MDA-MB-231, MDA-MB-468, and 4T1 cell lines. Cells were treated with 3 μM Nutlin-3A (N3) in panel (**d**). **f, g** Inactivation of the p53−p21 axis in MCF10A cells mimics the response of p53mt cancer cells. **f** MCF10A cells were transfected with siRNA to p53 or scramble-control siRNA, and then treated with 5 μM FdUrd for 6−48 h. **g** Cells were treated with 5 μM FdUrd or 1 μM doxorubicin 24−48 h, and then stained for RAD51 (red), γH2AX (green), and DNA (Hoechst, blue); images were taken at 600× magnification; scale bar 20 μm. Evaluation of γH2AX and RAD51 foci in response to 5 μM FdUrd in MCF10A and MDA-MB-231 cells (**h**, 6−48 h; **i**, 24−48 h). **h** The corrected total cellular fluorescence (CTCF) was determined using the γH2AX cellular fluorescence intensity from at least four random fields per group (>20 cells/group). Representative median scattered dot plots are shown. The comparison was made using the Log-rank test (***, $P < 0.001$; **, $P < 0.01$).

cells the RAD51 foci induction was transient and in fewer cells, indicating DNA break repair (Fig. 3i). Thus, the data indicated that inactivation of the p53−p21 axis in non-tumor cells recapitulate the DNA damage response to deoxyuridine analogues found in p53mt cancer cells, and p53mt TNBC cell lines accumulated DNA breaks in response to deoxyuridine analogues.

**Inhibition of PARP activates p53−p21 signaling.** Next, we questioned whether interference in the DNA repair process could further enhance a selective accumulation of DNA damage induced by deoxyuridine analogues in p53mt cells. The BER mechanism removes modified nucleotides, including ethynyl- and fluoro-deoxyuridine analogues, in cellular and mitochondrial DNA[45]. DNA glycosylases (e.g., UNG) recognize and excise damaged bases[46], initiating a repair process (Fig. 4a). The generated a-pyrimidinic sites (AP-sites) are then cleaved by AP-endonuclease APE introducing single-strand DNA breaks that are subsequently bound by poly (ADP-ribose) polymerase (PARP), which initiates recruitment of enzymes restoring the original DNA sequence (Fig. 4a). Inhibition of PARP interrupts DNA repair and can lead to DSBs and cell death[47].

First, we tested whether p53mt cells exhibit a differential response to PARP inhibitor (PARPi) compared to p53wt

cells. Treatment with PARPi, olaparib, inhibited poly-ADP-ribosylation (PARylation) activity at 100−1000 nM in both cell types (Fig. 4b). Notably, PARPi induced p53−p21 signaling in MCF10A cells, while p21 was not regulated in p53mt MDA-MB-231 cells (Fig. 4b). These findings were validated with a highly potent, but structurally different, PARPi talazoparib[48]. Talazoparib inhibited PARylation activity at 50 nM and effectively induced phosphorylation of Ser15-p53 in both cell lines irrespective of p53 status, whereas p21 induction was observed only in p53wt cells (Fig. 4c). The induction of the p53−p21 axis by PARP inhibitors was further validated in p53wt cell lines A549, CAL51, and WI-38 (Suppl. Fig. 6). PARP inhibitors did not induce DNA damage in p53wt and p53mt cells, based on the assessment of γH2AX levels (Fig. 4b, c and Suppl. Fig. 6). The cell-cycle data showed that PARPi increased G1 fraction in p53wt cells (Fig. 4d), consistent with activation of the p53−p21 axis (Fig. 4b, c). In p53mt cancer cells, both PARPi (olaparib and talazoparib) increased G2 fraction at the expense of the S phase, suggesting cell cycle arrest at G2 by PARPi in p53mt cancer cells (Fig. 4d). Thus, the data showed that PARPi activated the p53−p21 axis and increased G1 population in p53wt cells, while this response is compromised in p53mt cells, leading to G2 arrest.

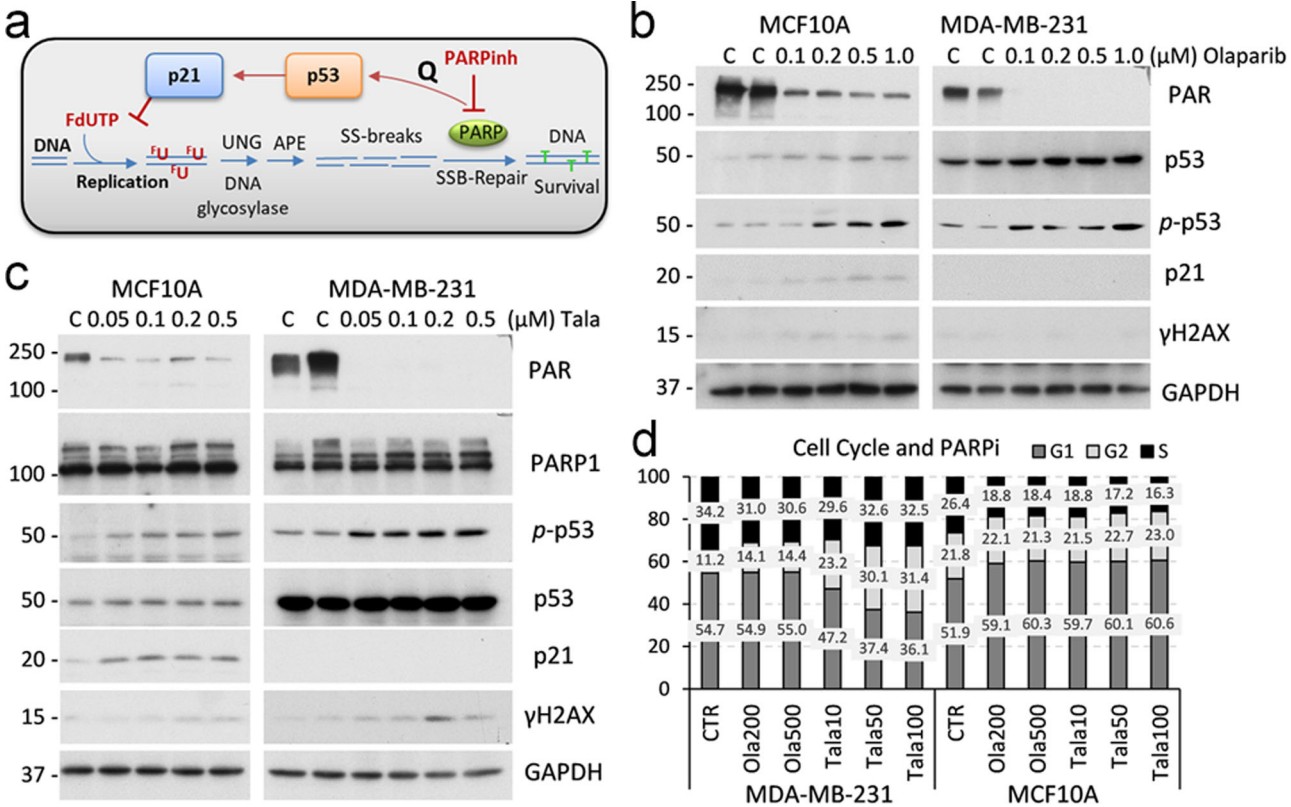

**Fig. 4 PARP1-inhibitor induces the p53−p21 axis and cell cycle arrest in p53wt cells. a** Schematic presentation of BER-mediated removal of the uracil analogue and the role of PARP1. PARP inhibition may activate p53. Immunoblots of whole-cell extracts from MCF10A (p53wt) and MDA-MB-231 (p53mt) cell lines treated with PARP inhibitors olaparib (**b**) or talazoparib (**c**) for 24 h at indicated concentrations. **d** Cell cycle data for MCF10A and MDA-MB-231 cells treated with PARPi, talazoparib (10, 50, and 100 nM), olaparib (200 and 500 nM), or vehicle control for 48 h.

**Inhibition of PARP enhances DNA damage in p53-mutant cancer cells.** Then, we examined whether PARP inhibition enhances DNA damage in response to the analogues in p53mt cells, while diminishing DNA damage in p53wt cells by inducing the p53−p21 axis. To test this idea, cells were treated with FdUrd alone or in combination with PARPi at the concentrations blocking the PARylation activity. In p53wt cell lines MCF10A and A549, PARPi alone increased p53−p21 signaling, but did not induce appreciable levels of γH2AX, while FdUrd activated the p53−p21 axis and induced a transient increase in γH2AX levels (Fig. 5a−d). The FdUrd-PARPi combination attenuated DNA damage response at 48 h in p53wt MCF10A, A549 (Fig. 5a−d), and EMT6 cell lines (Suppl. Fig. 7a). In p53mt cell lines (MDA-MB-231, MDA-MB-468, and BT549; all TNBC), FdUrd induced γH2AX, and PARPi strongly enhanced this response (Fig. 5b, c−e). As expected, neither treatment increased p21 in p53mt cell lines.

Next, the response to the drug combination was examined in p21ko MCF10A cell line. Talazoparib, FdUrd, or their combination markedly induced p53−p21 signaling in control MCF10A cells, while having a limited effect on γH2AX levels (Fig. 5f). In p21-deleted MCF10A cells, the drug combination induced γH2AX levels, while each drug alone activated p53 at the level comparable to control MCF10A cells (Fig. 5f).

Microscopic assessment of γH2AX foci confirmed the induction of DNA damage by the drug combination. FdUrd induced γH2AX foci in p53mt cells, while PARPi alone had a limited effect (Fig. 5g, h). The PARPi-FdUrd combination further increased γH2AX foci in p53mt but not in p53wt cells (Fig. 5g). Quantification of γH2AX fluorescence confirmed the accumulation of DNA damage in p53mt cancer cells compared to p53wt

cells (Fig. 5h). Together, the data indicated that PARP inhibition cooperates with deoxyuridine analogues in the induction of DNA damage in p53mt cells, while promoting p53−p21 signaling in non-tumor p53wt cells.

**Synergistic toxicity of deoxyuridine analogues and PARP inhibitors in p53mt cancer cells.** To determine the consequences of enhanced induction of DNA damage in p53mt cells by the FdUrd-PARPi combination, we examined whether PARPi and uridine analogues cooperate in the cytotoxicity responses. PARPi olaparib and talazoparib exhibited comparable IC50 values for MDA-MB-231 (17.63; 1.08 μM), MDA-MB-468 (17.27; 1.36 μM), and MCF10A (14.45; 1.59 μM) cell lines (Fig. 6a); these lines carry wild-type BRCA1/2 genes. The IC50 values markedly exceeded IC50 for BRCA1-deficient cell lines (<1 μM), and the inhibitory EC50 (1-10 nM) for PARylation activity in vitro[47] and in cell culture (Fig. 4, 50−100 nM). Cytotoxicity assays in combination with uridine analogues were done at the concentrations of PARPi that inhibit PARylation activity (Fig. 4b, c), but do not affect the growth of p53mt cells (Fig. 4d).

We found that PARPi olaparib sensitized p53mt cancer cell lines to FdUrd by nearly 10-fold (Fig. 6b and Suppl. Table 1). The combination index (CI-index) was 0.15 for MDA-MB-231 and CI = 0.16 for MDA-MB-468, indicating a synergistic interaction of FdUrd with PARPi. The drugs also synergized in mammary carcinoma p53-null 4T1 cells (Suppl. Table 1). In contrast, the drugs did not cooperate in p53wt cell lines (CI > 1.0): MCF10A, WI-38, EMT6, and CAL51 (Fig. 6b and Suppl. Table 1). Notably, 5-fluorouracil (5FU) did not cooperate with PARPi olaparib in any of the tested cell lines (Fig. 6c and Suppl. Table 2).

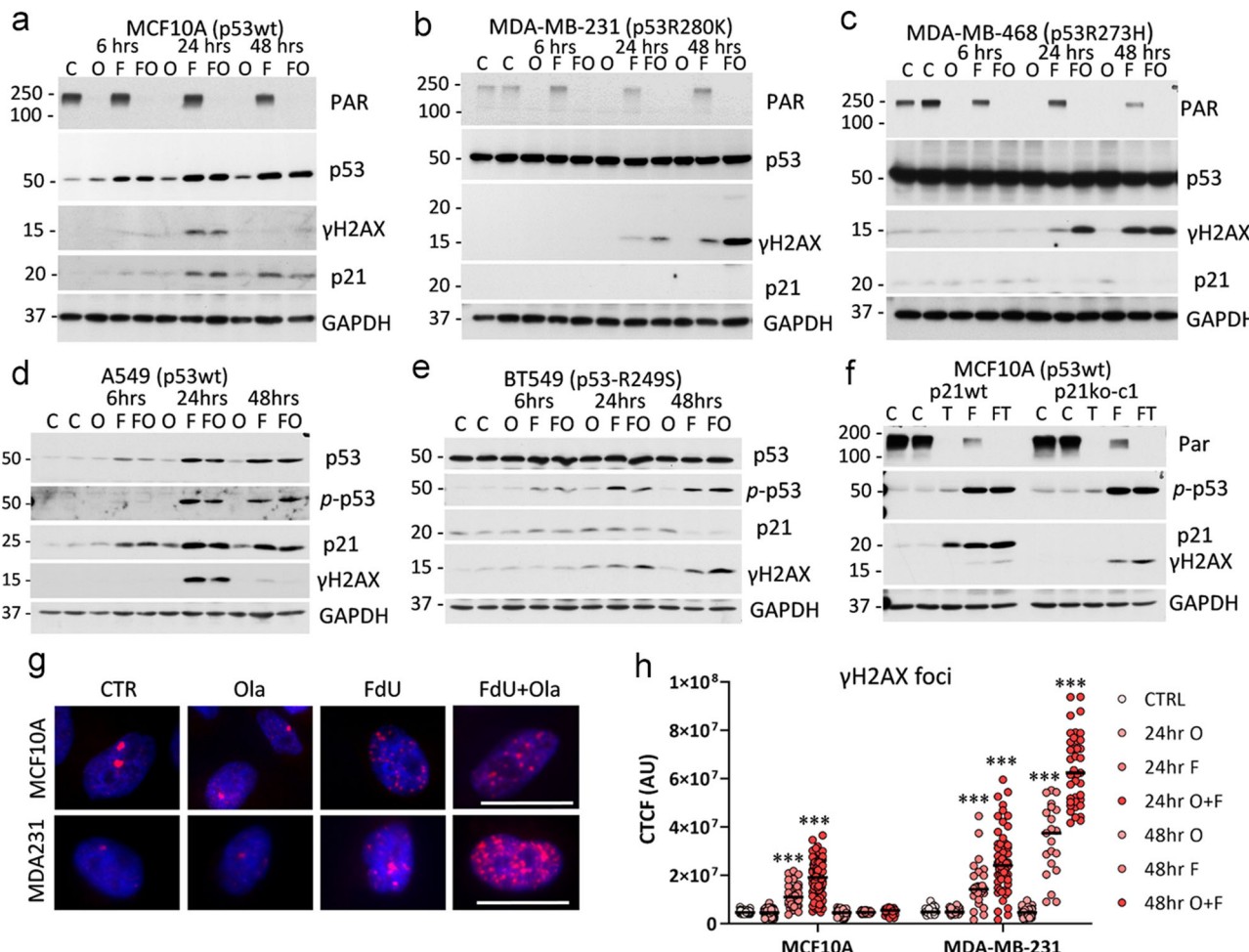

**Fig. 5 PARP1-inhibitor enhances DNA damage induced by FdUrd in cells deficient for p53–p21 signaling. a–e** Immunoblots of whole-cell extracts of p53wt (MCF10A; A549) and p53mt (MDA-MB-231, MDA-MB-468, and BT549), cell lines treated with 0.5 μM olaparib (Ola, O), 5 μM floxuridine (FdUrd, F), or their combination (FO). **f** Control and p21-deleted MCF10A cells were treated with 100 nM talazoparib (T), 5 μM FdUrd (F), or their combinations (FT). **g, h** Detection of γH2AX foci in cells treated as described in (**a**). Cells were stained for γH2AX (red) and DNA (Hoechst, blue), and images were taken at 600× magnification, scale bar 20 μm. The corrected total cellular fluorescence (CTCF) was determined using γH2AX fluorescence of at least 30 cells/group. Representative median scattered dot plots are shown. The comparison was made using the Log-rank test (***, $P < 0.001$).

The synergism of FdUrd with PARPi was further validated using PARPi talazoparib. We found a strong cooperativity between FdUrd and talazoparib in p53mt cell lines (CI < 0.2), while no cooperativity was observed in MCF10A cells (Fig. 6e, CI = 1.07). Isobologram data indicated a strong cooperativity between FdUrd and PARPi in p53mt cells, based on the position of all experimental values below the 0.75 cutoff line (Fig. 6f). Next, we examined whether PARPi cooperates with TAS102, a new anticancer drug consisting of trifluoro-thymidine (TFT) and tipiracil[49]. The assays showed a strong cooperative interaction between TAS102 and talazoparib in p53mt cell lines (CI < 0.2), while cooperativity was not observed in p53wt MCF10A cell line (Fig. 6g; CI = 0.99). TAS102 also cooperated with olaparib in p53mt cell lines but not in p53wt cells (Suppl. Table 3). We further confirmed by immunoblotting that TAS102 induced DNA damage (γH2AX) in MDA-MB-231 cells (Suppl. Fig. 7b).

To assess the toxic effects of the drug combination, we evaluated the activity of caspase-3/7 in cell lysates. Treatment of p53mt MDA-MB-231 cells with a combination of FdUrd and olaparib markedly stimulated caspase-3/7 activity compared to control and each drug alone (Suppl. Fig. 7c). Likewise, the TAS102-talazoparib combination markedly increased caspase-3/7 activity in p53mt cells, whereas there was no induction in p53wt

MCF10A cells (Suppl. Fig. 7d). Thus, the data demonstrated that PARPi selectively sensitizes p53mt cancer cells to the cytotoxic effects of deoxyuridine analogues, while reducing their toxicity in non-tumor p53wt cells.

**Inhibition of PARP enhances the anti-tumor activity of TAS102 in p53-mutant TNBC model.** The efficacy of the TAS102/PARPi combination was examined in a mouse breast cancer model with MDA-MB-231 cells representing TNBC. Cells were implanted into the mammary gland of SCID mice, and treatments were initiated once the primary tumor reached 100mm³. PARPi alone did not affect tumor growth compared to the vehicle control group (Fig. 7a). This result was consistent with the absence of genetic alterations in BRCA1/2 and other HR repair genes, based on genomic data. Treatment with TAS102 alone reduced tumor growth while mouse weight was not reduced, indicating that the treatment was well-tolerated (Fig. 7a, b). Notably, the TAS102-olaparib combination further reduced tumor growth compared to TAS102 alone (Fig. 7a), while the drug combination was well-tolerated without major changes in mouse weight (Fig. 7b). Notably, the drug combination significantly increased cell death (active caspase-3) in tumor xenografts compared to each drug alone (Fig. 7c, $P < 0.01$). Evaluation

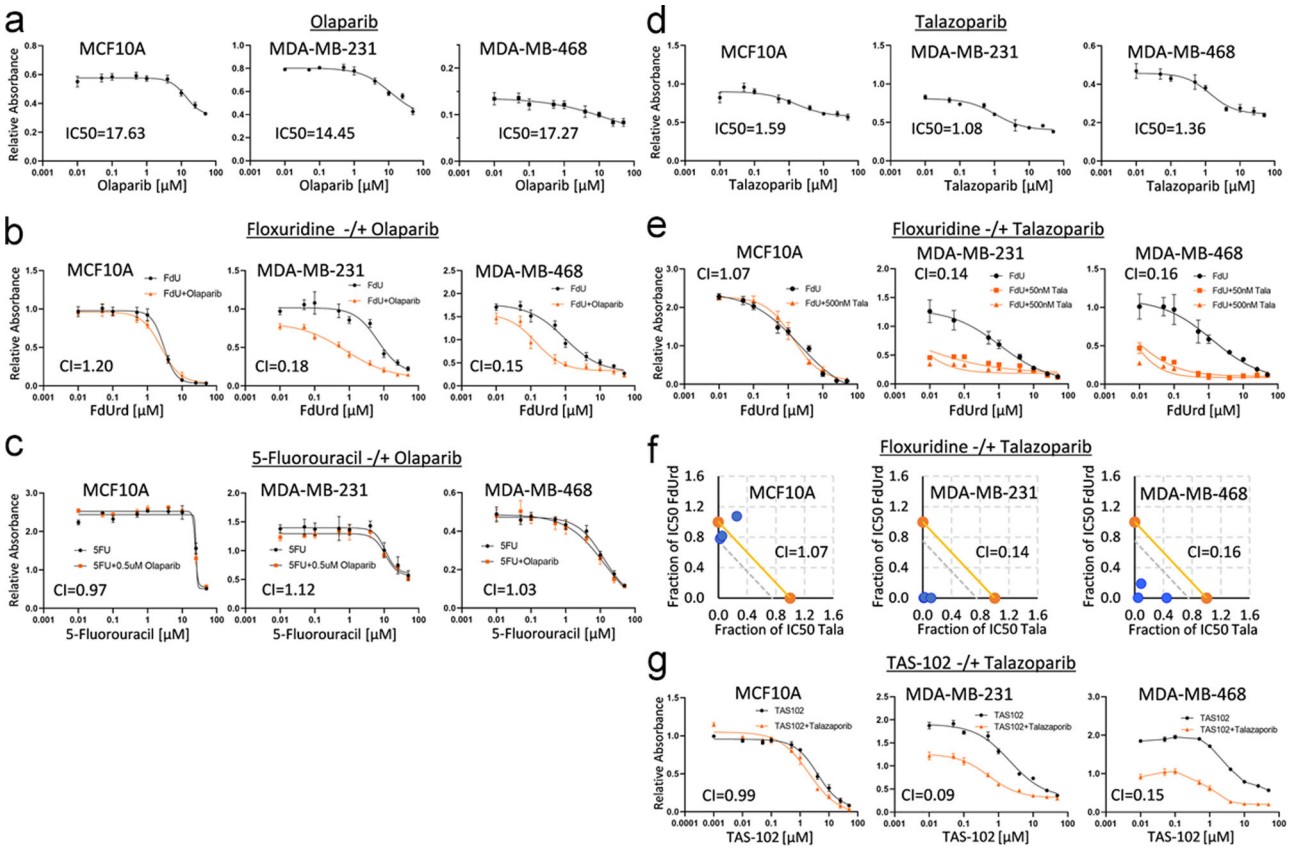

**Fig. 6 PARP1-inhibitor enhances cytotoxicity of deoxyuridine analogues in p53-mutant tumor cells. a, d** Cytotoxicity curves for PARP inhibitors (PARPi) olaparib and talazoparib. **b** Cytotoxicity curves for floxuridine (FdU) alone and in the presence of 0.5 μM olaparib. **c** PARPi olaparib (0.5 μM) does not enhance the cytotoxicity of 5-fluoro-uracil (5FU) in p53wt and p53mt cells. **e** PARPi talazoparib (100 nM) enhances the cytotoxicity of floxuridine (FdU) in p53-mutant cancer cell lines (MDA-MB-231; MDA-MB-468). **f** Isobolograms for floxuridine (*Y*-axis) and talazoparib (*X*-axis) and mean combinatory index (CI-index). **g** Cytotoxicity of TAS102 (5-trifluoro-thymidine and tipiracil) is enhanced by PARPi talazoparib (100 nM) in p53mt cancer cell lines, CI < 0.20. Assays were repeated at least two times in six replicates for cytotoxicity.

of survival, based on time-to-arrive at 300 mm³ tumor volume, showed a statistically significant improvement ($P < 0.01$) in the combination group compared to TAS102 alone (Fig. 7d). The immunohistochemistry data confirmed the uptake of TAS102 by tumor cells (Suppl. Fig. 8a, BrdU). The combination therapy increased DNA damage (γH2AX) and decreased tumor cell proliferation (Ki67) (Suppl. Fig. 8a). Histological inspection showed pulmonary metastases in the control, olaparib-alone, and TAS102-alone groups, while the combination-treated mice did not have metastases (Suppl. Fig. 8b). These findings demonstrated that the drug combination was more effective in the reduction of primary tumor growth and metastases to the lungs than each drug alone.

## Discussion
Management of advanced breast cancers is a major clinical problem with limited therapeutic options. Although breast cancers commonly carry genetic alterations in p53[21], only a small number of therapeutic strategies target this genetic abnormality[3,4,16]. Here, we identified a selective inducer-amplifier strategy for effective targeting p53 mutant cancer (Fig. 7e). This synthetic lethality strategy was validated in preclinical models using clinical drugs that have never been combined before. We found that p53mt cancer cells exhibit dysregulation in BER-mediated DNA repair, resulting in the accumulation of DNA damage in response to deoxyuridine analogues. Further work showed that PARP inhibitors cooperate with deoxyuridine analogues to enhance

DNA damage in p53mt cells, whereas wild-type p53 carriers respond with activation of p53−p21 signaling and cell-cycle arrest (Fig. 7e). The anticancer synergy of the drug combination was confirmed in preclinical cancer models, with no major overall toxicity in mice.

Genomic data demonstrated high expression of RRGs in TNBC, which is consistent with the expression of the periodic cell-cycle genes[50]. Recent studies showed that TNBCs also exhibit the elevated activity of ribosome biogenesis[51]. Both cellular processes consume substantial nucleotide resources that may imbalance the pool of nucleotides, triggering replication stress and contributing to mutational burden[22–26]. In addition, mutant p53 may directly affect nucleotide metabolism[52]. Consistent with these notions, TNBCs show high mutation frequencies with a prevalence of C-to-T transition, which is typically associated with misincorporation or modification of nucleotides[19,20]. These DNA lesions are normally removed by DNA repair mechanisms such as MMR and BER[28,29]. Genomic data showed that BER and MMR genes are highly expressed in p53mt cancers, including TNBCs, while their genetic alterations are rather rare events. Furthermore, the BER/MMR expression levels tightly correlated with the expression of RRGs. Conceivably, activation of DNA repair genes may reflect the loss of p53 function, which is critical for the control of cell-cycle in response to DNA damage[15]. On the other hand, p53 may directly regulate BER activity[39,53] and promote cell death if DNA damage is unrepairable[15].

Here, we found that p53mt cancer cells exhibit dysregulation in BER-mediated repair, in part due to the inability to interrupt

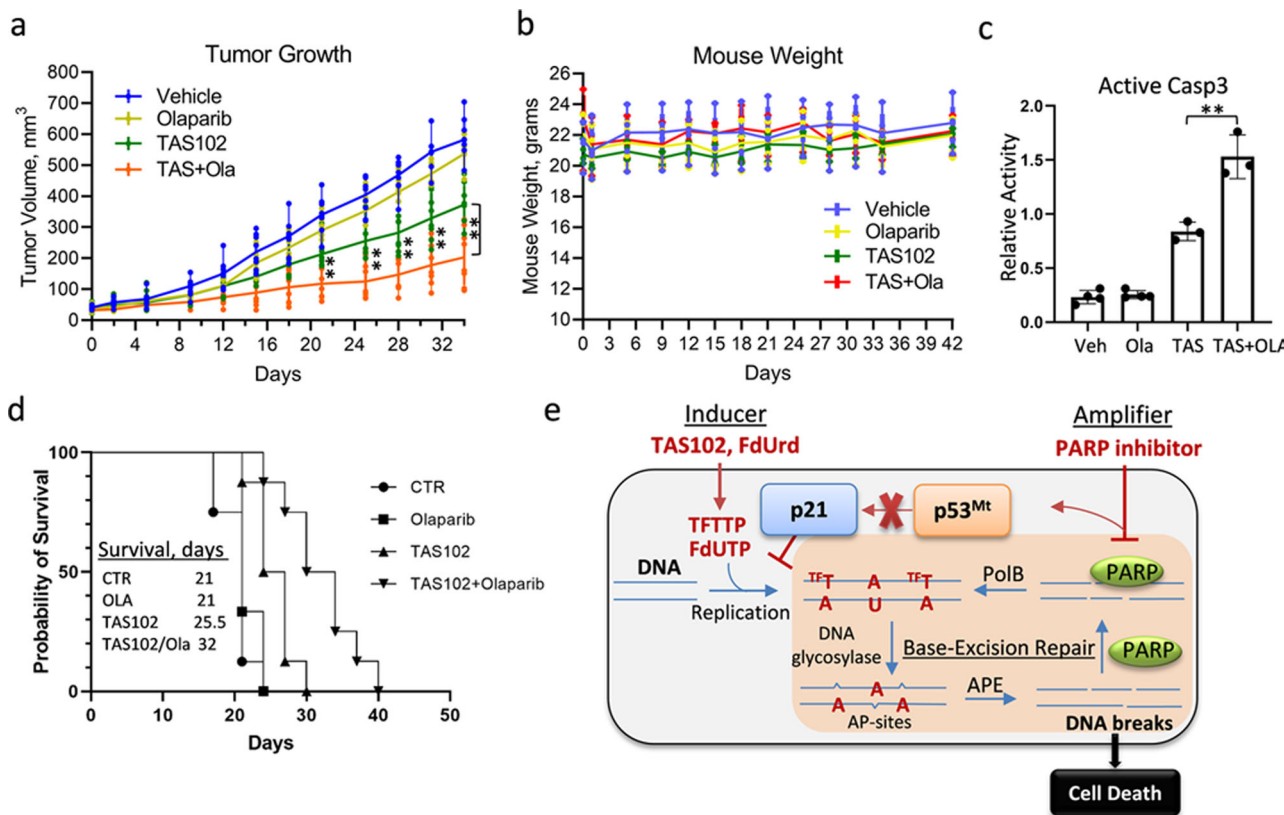

**Fig. 7 Tumor growth and metastasis are reduced by systemic treatment with a combination of TAS102 and PARP1-inhibitor olaparib. a** Breast cancer MDA-MB-231 cells were orthotopically implanted into female BALB/c mice. At tumor size 100 mm³, mice were randomly divided into four groups and treated by daily oral gavage with vehicle control, olaparib alone (50 mg/kg), TAS102 alone (50 mg/kg), or TAS102 + olaparib in combination (50 mg/kg each drug) on schedule 5 days-on, 2 days-off. Tumor size was measured two times per week. The comparison was made using the Log-rank test (*, $P <$ 0.05). **b** Mouse weight was measured twice weekly. **c** Active caspase-3 by immunohistochemistry. **d** Survival was evaluated using Kaplan−Meier estimator based on time-to-arrive at 300 mm³ of tumor size. Median survival 21 days (vehicle and olaparib), 25.5 days (TAS102), 32 days (TAS102 + olaparib). The comparison was made using the Log-rank test ($P <$ 0.01). **e** Model of the cooperative interaction of deoxyuridine analogue (e.g., TAS102 containing trifluoromethyl-deoxyuridine (TFT/TFdU) and tipiracil) and a PARP inhibitor (i.e., olaparib or talazoparib) in selective damaging p53mt cancer cells.

DNA replication. In response to the EdUrd pulse, p53mt cells proceeded in the S phase and accumulated DNA damage, whereas p53wt cells responded with G1 arrest and repaired DNA lesion, based on the cell cycle data and γH2AX levels. Our data support a critical role of the p21 CDK inhibitor, a p53 target, in these actions since p21-deleted p53wt cells exhibited a response similar to p53mt cancer cells, demonstrated by the γH2AX data in response to deoxyuridine analogues. This finding is consistent with p21 function as a negative regulator of PCNA-dependent DNA replication[40]. Thus, our data indicate a major role of p21 in the control of DNA damage response by interrupting replication and allowing successful DNA repair, while p53 may also regulate DNA repair, including BER, through multiple direct and indirect mechanisms[15,17].

Our data demonstrate that fluorinated and ethynyl-deoxyuridine analogues selectively induced DNA damage in p53mt cancer cells. Fluorinated uridine antimetabolites (FdUrd and 5FU) have been in clinical practice for several decades and their pharmacology is well studied[54,55]. The main mechanism of action for these analogues involves inhibition of thymidylate synthase (TS) by a common metabolite FdUMP, leading to a reduction of dTTP levels and promoting the incorporation of dUTP and FdUTP into the genome[46,55]. The major difference between the two analogues is a preferential incorporation of 5FU into RNA while FdUrd is mainly incorporated into DNA[46]. Genomic fluorouracil (FU) forms A-U pairs and FU subsequently

is removed by DNA glycosylases (UNG, TDG, and SMUG1), initiating BER (Fig. 7e). The repair process leads to the incorporation of U and FU into DNA by polβ and to a new round of BER, resulting in multiple futile repair cycles[33]. Similarly to FdUrd, EdUrd, and TFT (5-trifluoro-2′-deoxythymidine, a component of TAS102) also inhibit TS (0.38 nM for TFT) and are incorporated into DNA, although TFT shows a prolonged retention in DNA[56]. In p53wt normal cells, deoxyuridine analogues induce DDR and activate the p53−p21 axis, halting DNA replication, while p53mt cells do not stop DNA replication leading to the accumulation of DNA breaks (see above). This scenario was evident by a transient increase of γH2AX, RAD51 foci, and G1 arrest in p53wt cells, and accumulation of γH2AX and RAD51 foci in p53mt cell lines. The data argue that inactivation of the p53−p21 axis underscores the differential response to deoxyuridine analogues in p53mt cells, indicating the importance of p21-mediated inhibition of replication in the effective DNA repair.

PARP inhibitors olaparib and talazoparib enhanced DNA damage and cytotoxicity caused by deoxyuridine analogues in p53mt cells, while activating the p53−p21 axis and increasing G1 population in p53wt cells (this was validated in both human and mouse cell lines). The anti-tumor synergism was observed at amounts that inhibited PARP activity, but was not toxic to cells. These PARP inhibitors trap PARP protein at single-strand DNA breaks, causing replication fork stalling or collapse, and leading to

the formation of one-ended DNA DSBs[47,48]. HR-deficient cancer cells, i.e., BRCA1/2 mutants, are incapable of faithfully repairing such DNA lesions, resulting in cancer cell death[11,57]. Thus, the mechanistic explanation of synergistic toxicity of PARPi and deoxyuridine analogues in p53mt cancer cells is likely associated with the accumulation of DNA DSBs due to PARP-trapping activity. Consistent with the PARP-trapping model, PARPi markedly enhanced DNA damage and cancer cell death when given in combination with TAS102 or FdUrd in vitro (Suppl. Fig. 7c, d) and in vivo (Fig. 7c). In this regard, TAS102 alone induces minimal double-strand breaks[58] and shows only moderate effects in the clinical trial[59]. Hence, the TAS102-PARPi combination may improve the anti-tumor activity of TAS102 while protecting normal tissues.

Our work illustrates a new concept that utilizes the inducer-amplifier strategy to achieve selective synthetic damage to cancer cells (Fig. 7e), while limiting the impact on non-tumor tissues. Clinical application of PARP inhibitors (olaparib and talazoparib) as monotherapy is limited to HR-deficient cancers representing a small proportion (<5%) of all breast cancer cases[13]. A new drug regimen combining PARP inhibitors and deoxyuridine agents, i.e., TAS102 or FdUrd, expands the clinical utility of these therapeutic agents to p53-mutant cancers that account for the majority of TNBC and nearly half of all BC cases. Importantly, our work may have broader implications, since nearly half of all human cancers carry p53 mutations.

## Methods

**Cell lines and culture conditions.** All human cell lines were authenticated using short tandem repeat profiling by ATCC or the Roswell Park Core within the last three years. All studies were mycoplasma-free cells. Human non-tumor breast epithelial MCF10A (RRID: CVCL_0598), metastatic breast cancer MDA-MB-231 (RRID:CVCL_0062), MDA-MB-468 (RRID: CVCL_0419) and BT549 (RRID:CVCL_1092), breast cancer CAL51 (RRID:CVCL_1110), lung adenocarcinoma A549 (RRID:CVCL_0023), human embryonic fibroblast WI-38 (RRID: CVCL_0579) and mouse mammary carcinoma EMT6 (RRID:CVCL_1923) and 4T1 (RRID:CVCL_0125) cell lines were obtained from American Type Culture Collection (ATCC, Manassass, VA, USA), and cultured as recommended by ATCC. Human breast epithelial MCF10A-p21ko cell lines were from Dr. Ben Ho Park and are described elsewhere[43].

**PARP inhibitors, antibodies, and other reagents.** Details on reagents and antibodies can be found in Supplementary Information.

**Mice.** Female SCID/CB17 mice (6−7-week-old) were obtained from a colony of SCID/CB17 mice bred and maintained at the Animal Facility of the Roswell Park Comprehensive Cancer Center (RP). Animals were kept in microinsulator units and were provided with food and water ad libitum according to a protocol and guidelines approved by the Institute Animal Care and Use Committee (IACUC). The facility is certified by the American Association for Accreditation of Laboratory Animal Care (AAALAC) and is in accordance with current regulation and standards of the US Department of Agriculture and the US Department of Health and Human Services.

Mice were inoculated into the 4th mammary fat pad with exponentially growing MDA-MB-231 tumor cells ($1\times10^{6}$/mouse). Tumor growth was monitored by measuring tumor diameters with electronic calipers twice/week. Volumes were calculated using the formula (length) × (width)$^2$/2. Mouse weights were measured twice/week. Once the tumor volume reached 100 mm$^3$, the mice were randomly divided into four groups: vehicle control, olaparib, TAS102, and TAS102 + olaparib ($n = 8$ mice/group). The PARP inhibitor olaparib and TAS102 were dissolved in 12% HPCD, (2-hydroxypropyl)-β-cyclodextrin, in Dulbecco's phosphate-buffered saline (DPBS). Drugs were given at 50 mg/kg by oral gavage on schedule 5-days-on and 2-days-off. At the endpoint, the mice were euthanized and subjected to necropsy and organ collection. Tumor tissues were collected for RNA and protein analyses by snap-freezing in liquid nitrogen. Blood was collected for CBC by cardiac puncture.

**Complete blood counts.** At the endpoint, blood was collected by cardiac puncture into EDTA-containing tubes to prevent coagulation. Analysis was performed using the HemaTrue Analyzer and HeskaView Integrated Software version 2.5.2.

**Cytotoxicity assay.** Cells were plated at a density of 5,000 cells/well in a 96 well-plate and then treated in six replicates with the appropriate drugs at varying amounts for 24 h. Media was replenished with media with or without PARP inhibitors and cells were incubated for 96 h. Cells were stained with 1% Methylene Blue for 30 min, rinsed with water, dried, and then solubilized in 5% SDS in PBS, and read at 650 nm. IC$_{50}$ values were generated using GraphPad Prism8 (Version 8.4.2).

**Immunoblot assays.** Whole-cell lysates were prepared using NP40 Lysis Buffer (0.88% NP-40, 132 mM NaCl, 44 mM Hepes, and 8.8 mM NaF) supplemented with 2 mM sodium orthovanadate, 1 mM PMSF, and Protease Inhibitor Cocktail. Protein concentrations were measured using the Bio-Rad DC Protein Assay. Proteins were resolved on SDS-PAGE and transferred to nitrocellulose membranes. Protein bands were visualized using ECL chemiluminescent reagent. Changes in protein levels were quantified using ImageJ software version 1.52 C and normalized to GAPDH.

**Immunohistochemistry (IHC).** Tumors and organ tissues were fixed in 10% (v/v) formalin, before embedding in paraffin by the Pathology Core. H&E and other stains were carried out by the Pathology Core as described in ref. [60]. Details of antibodies and reagents, and expanded methodology for immunohistochemistry, blood vessel, and Ki-67 index evaluation, and statistical analysis can be found in the Supplementary Information. Briefly, for active caspase-3 activity, tumor xenografts were fixed in 4% paraformaldehyde and embedded into paraffin blocks. Staining with cleaved caspase-3 antibody was performed by the Roswell Park Comprehensive Cancer Center (RPCCC) Pathology Core. Slides were scanned into Aperio ImageScope version 12.4.0.5043. Images from individual tumor sections were recorded at ×40 magnification and analyzed using ImageJ software version 1.52 C.

**Flow cytometry.** All samples were analyzed on an LSRFortessa Cytometer (BD Biosciences) running FACSDiva (Version 6.1.3), and the data were processed using FCS Express 7 (Version 7.04.0016). For the EdUrd pulse experiments, 300,000 cells per well were seeded in a six-well plate, and the following day media was replaced with base media containing 5% dialyzed FBS. Cells were then incubated with 10 μM EdUrd for 2 h, while untreated cells served as the negative control. Following the 2-h pulse, the cells were washed twice with DPBS and the media was replenished. Collection of cells began at $t = 0$ h up to $t = 72$ h post EdUrd-pulse. Cells were collected using standard trypsinization, washed in 1% BSA in DPBS, and fixed in 4% paraformaldehyde for 15 min. Cells underwent two more washes in 1% BSA/DPBS before being permeabilized in 1X saponin buffer. To label the incorporated EdU, cells were subjected to "click-it" reaction with Cu(II)SO$_4$, Tris-HCl, pH 8.5, THTPA, ascorbic acid, and either Cy3 azide or AFDye 488 azide for 30 min. DNA content was labeled with either Hoechst 33342 or propidium iodide containing RNase A. All samples were washed in 1% BSA/DPBS, resuspended in 1X saponin buffer, and transferred to polystyrene tubes. Experiments were repeated three times and representative histograms and dot-plots shown. For cell cycle analysis, cells were seeded at 300,000 cells/well in six-well plates and then treated with various amounts of olaparib and talazoparib for 24 h. Collected cells were fixed for 24 h in ice-cold 70% ethanol and stained for 2 h at 4 ˚C in Krishan DNA Buffer (propidium iodide, sodium citrate, RNase A, NP40, and 0.1 mM HCl). Samples were sorted using a BD LSRFortessa cytometer running FACSDiva (Version 6.1.3), and the data were analyzed using ModFit Lt software (Version 5.0.9). Experiments were repeated twice with representative histograms shown.

**Immunofluorescence microscopy.** For evaluation of genomic EdU, cells were grown on glass coverslips, and then pulse-labeled with EdUrd as described for flow cytometry. Cells were fixed at various times with 4% PFA and permeabilized with 0.05% Triton X-100, and then EdU was labeled with Cy3 azide using "click-it" reaction as described above. DNA was labeled with Hoechst 33342 before mounting on glass slides. Fluorescence images were taken with a Plan Apochromat 60×/1.40 NA oil objective using Nikon TE2000-E inverted microscope equipped with a CoolSNAP HQ camera. The images were acquired using MetaVue software (v7.7.3, Molecular Devices). The experiments were repeated twice with representative images shown. For the evaluation of γH2AX foci, cells were grown as above and treated alone or in combination with 5 μM FdUrd and 0.5 μM olaparib for various times, then fixed as described above. Samples were blocked with 3% milk in PBS at room temperature (RT), and then incubated with γH2AX antibody (1:400) in 1% milk/PBS followed by incubation with Texas red-conjugated secondary antibody (1:500). DNA was labeled with Hoechst 33342 before mounting on glass slides. Fluorescence images were acquired as described above. The level of γH2AX cellular fluorescence intensity was determined as the corrected total cellular fluorescence (CTCF). Briefly, a minimum of four random fields per treatment group were evaluated using MetaVue imaging software (Version 7.7.3, Molecular Devices). Three background readings were measured for each field of view and individual cells that were positively stained with γH2AX were selected with a region freehand tool. For each treatment group >40 cells were evaluated and the CTCF values were calculated using the following formula: integrated density − (area of cell × average background fluorescence). Data analysis was performed

using Microsoft Excel and representative median scattered dot plots were generated in GraphPad Prism 8 (Version 8.4.2).

**Caspase-3/7 activity**. Cells were seeded in 96-well plates at 5,000 cells/well and incubated at 37 °C, 10% $CO_2$. Treatments in triplicates were given the next day at the indicated concentrations and time. Caspase-3/7 activity was measured using Promega Caspase-Glo 3/7 Assay following the manufacturer's protocol. Luminescence was measured in the VERITAS microplate luminometer with version 1.9.3 software, presented as relative luminescence units (RLU), and quantified with Microsoft Excel.

**Metadata analysis**. Heat-map of expression profiles was generated using the TCGA Breast Cancer dataset, Project ID: TCGA-BRCA, dbGaP Study Accession: phs000178. Expression Z-scores and mutation data were downloaded *via* the cBioPortal tool https://www.cbioportal.org/.

Gene lists for cell-cycle-related genes are generated using Cyclebase_3.0 database http://www.cyclebase.org[61]. DNA repair gene lists were derived from the KEGG database http://www.genome.jp/kegg/[62]. Plots throughout are the sample means ± 1sd. Expression of DNA repair and RRGs in MDA-MB-231 and MCF10A were derived from gene expression profiles reported previously[51]. Heatmaps of differential genes were drawn by using the R-package, ComplexHeatmap. All the data were analyzed and processed in R/Rstudio 4.0.3 version; the data are available in the Supplementary Data 1 files.

**Statistics and reproducibility**. Statistical significance of data comparisons was determined using the Student's unpaired *t*-test with a two-tailed distribution. Statistical significance was achieved when $P < 0.05$. All experiments were performed for at least two times. Survival was evaluated using the Kaplan−Meier estimator with the log-rank test, based on time-to-arrive at a tumor volume of 1 $cm^3$ using GraphPad Prism 8.

**Reporting summary**. Further information on research design is available in the Nature Research Reporting Summary linked to this article.

## Data availability

The heat-map data that support the findings of this study are generated using the cBIoportal for Cancer Genomics at http://www.cbioportal.org/ and the breast cancer TCGA dataset (Project ID: TCGA-BRCA; dbGaP Study Accession: phs000178) available in a public repository the Genomic Data Commons Data Portal at https://portal.gdc.cancer.gov/. All the other data supporting the findings of this study are available within the article, the Supplementary Data 1 files, the Supplementary Data 2, and Supplementary Information (Supplementary Figures, Supplementary Tables, and Supplementary Figures for uncropped blots).

## Code availability

The raw data and R-code files employed for the generation of Fig. 1 and Suppl. Fig. 1 data are deposited in Supplementary Data 1, the R-code is available at https://github.com/Bakin-lab/BRCA-BER.

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

## Acknowledgements
We gratefully acknowledge the generous help from Flow and Image Cytometry Facility, the Pathology Resource Network, Preclinical Imaging Services, and Laboratory Animal Resource (LAR). We thank Drs. Elizabeth Repasky and David Goodrich for discussion of the manuscript. Department of Defense BCRP Program BC151886 (to AVB), Roswell Park Alliance Foundation (to AVB), Metastatic Breast Cancer METAvivor foundation (to AVB), NIH grant AI128421 (to TM), and in part by NIH R25CA181003 (to MM) and the Roswell Park Comprehensive Cancer Center Support Grant, P30CA016056

## Author contributions
A.V.B. conceived the idea, A.V.B and J.Z. designed the experiments, J.Z., B.S., M.W., M.M., M.A., and K.E. performed the experiments and analyzed the data; A.V.B. and C.F. analyzed and interpreted the data; T.M. helped in the interpretation of the data; K.E. helped in the analysis and interpretation of the TCGA genomic data; A.V.B., K.E, B.H.P., R.I., C.F., T.M., and J.Z. contributed to preparing and writing the paper.

## Ethics declarations
A study protocol and guidelines approved by the Institute Animal Care and Use Committee (IACUC). The facility is certified by the American Association for Accreditation of Laboratory Animal Care (AAALAC) and in accordance with current regulation and standards of the US Department of Agriculture and the US Department of Health and Human Services.

### Competing interests
The authors declare no competing interests.
