## [Peer Review File · Communications Biology]

Reviewers' Comments:

Reviewer #1:

Remarks to the Author:

In this manuscript, entitled "Selective therapeutic strategy for p53-deficient cancer by targeting dysregulation in DNA repair", Zonneville et al identify a DNA repair vulnerability in p53-mutant breast cancers and exploit this genetic liability for therapeutic purposes. The authors showed that basal/triple-negative breast cancers (TNBCs) present higher expression of base excision repair (BER) compared to other subtypes of breast cancer. In addition, they demonstrated that fluorinated pyrimidine analogues induce DNA damage that persist in p53-mutant TNBC cells, while it is repaired in p53 functional cells. Thereby, the combination of fluorinated pyrimidine analogs and PARP1 inhibitor, which target the BER pathway, induced cellular cytotoxicity and suppressed tumor growth in mice bearing p53-mutant TNBC cells.

This is an interesting work with an appropriate experimental design to test the proposed hypothesis. The in vitro experiments support the rationale of the study and the in vivo data highlights the therapeutic potential of targeting the BER pathway in p53-mutant TNBC. The manuscript is well written and the scope is of interest for the cancer community readership. I think that the conclusions are well supported; however, some major points should be addressed.

Major points

1. The authors use a non-tumoral p53 wild-type (wt) breast epithelial cell line, MCF10A, to compare outcomes from p53-mutant breast cancer cell lines in response to the indicated treatments. This is of great value to show that the combination therapy preferentially targets p53-mutant TNBC cells over healthy cells; however, to reinforce that targeting the BER pathway is a selective therapy for p53-mutant TNBC, the work would benefit from showing at least in vitro the effect (cytotoxicity profiles, tumor growth) of treating p53-wt TNBC cells (i.e. SUM102PT cells) with PARPi and fluorinated pyrimidine analogs.
2. Accumulation of unrepaired EdUrd/FdUrd-induced DNA damage in p53-mutant breast cancer cells could lead to p53-independent cell death. The work would benefit from showing the apoptotic profiles (i.e. annexin V stain, cleaved caspase detection by western blotting) of EdUrd/FdUrd-treated MCF10A and MDA-MB-231 cells after drug wash out and during continuous treatment.
3. The role of p53 in enabling repair of FdUrd-induced DNA damage is evaluated at 24 and 48h. Likewise, p21 should also be assessed at 48h for DNA damage repair in p21-wt and p21-knockout cells.
4. Combination of TAS-102 and Olaparib leads to DNA damage accumulation (increased gH2AX) and reduced proliferation (decreased Ki67 levels) in p53-mutant tumors, observed by histological analyses. The work would benefit from showing also cell death (i.e. cleaved caspase 3) in the treated xenografts.

Minor points

1. Figure 6A. Cytotoxicity curve should be shown for Olaparib alone in MDA-MB-468 for comparison, since combined treatment with Olaparib and fluorinated pyrimidine analogues (FdUrd or 5-FU) is used in this cell line.
Figure 6F. Cytotoxicity curves for Talazoparib alone should be shown in MCF10A, MDA-MB-231 and MDA-MB-468 for comparison.
2. Lines 335-338. Attenuation of p53-p21 signalling and gH2AX levels in p53-wt cells in response to

PARPi-FdUrd combination is not well appreciated in blots of Figure 5A, D and Supplementary Figure 3C. Please, provide a longer exposure to clearly show the differences if possible.

3. Figure 1A-B. For clarity, please, add the X axis line in (A) and Y axis labels (i.e. "S-phase genes expression, fold change..."; "tumor mutation burden, mutations/Mb") in (A-B).

Supplementary Figure 1. Change (E) to (C) in the figure legend.

Supplementary Figure 1C. Please, add Y axis label.

Supplementary Fig 1C. It is not clear to the reader if panel (C) shows gene expression levels for MDA-MB-231 or MCF10A, or if it shows the fold change of gene expression levels of MDA-MB-231 over MCF10A. I would recommend for clarity and comparison purposes to represent gene expression levels of the two cell lines in separate graphs.

Figure 2A, C. DNA content (Hoechst) should be represented in linear scale.

Figure 3F. Please, indicate concentration and hours of FdUrd treatment in the figure legend.

Line 274 "EdUrd treatment induced activation and stabilization of p53 irrespective of p53 status...". Stabilization of p53 is not observed in p53-mutant MDA-MB-231 and MDA-MB-468 cells (Figure 3A-C). Please, change sentence accordingly.

Figure 4D. For clarity, please, indicate Olaparib and Talazoparib concentrations, instead of a concentration range, in panel (D) and in the figure legend.

Figures 3F and 5F. Add scale bars to microscopy images.

Figure 5F. Indicate hours of treatment in the figure legend.

Figure 6A-D, F. Axis labels and legend in panels are non-readable, please, replaced this figure by a version with better resolution.

Figure 6E. Add axis labels and legend in panel (E).

Figure 6. Remove panel (G) mention in the figure legend. There is no panel (G) in the actual figure or in the text.

Supplementary Figure 3. Panel (C) is referred in the text (line 338) earlier than panel (B) (line 381). Please, format supplementary figure for consistency with the text.

4. Figure 3E. p21wt MCF10A cells are treated with FdUrd 5 μ M for 24h, as in Figure 3B and 3D (siRNA-SCR); however, upregulation of gH2AX levels at this timepoint is not observed, in contrast to what previously shown. Please, show a more representative blot of this experiment in line with previous observations or explain in the text why gH2AX levels are not increased in this case.

5. Figures 3F. Could the authors explain the discrepancy in nuclear size of CTRL vs FdUrd-treated MDA-MB-231 cells? Please, show a more representative image of CTRL MDA-MB-231 cells.

6. Levels of Olaparib-induced p21 activation are inconsistent across experiments at the same timepoint (24h) and concentration of drug used (0.5 μ M; Figure 4B and 5A); p21 protein upregulation is minimal in MCF10A cells from Figure 5A. To help follow the rationale of the work better, I would suggest to provide blots that reflects more accurately and consistently Olaparib-induced p21 activation across experiments.

7. Previous works (i.e. PMID: 25700705) have shown that TAS-102 induces p53-p21 signalling with minimal double strand breaks, please, contextualize findings in line with literature.

8. Supplementary Figure 3A. p53 stabilization and p21 upregulation in Olaparib-treated WI-38 (p53 wt) cells is not very clear in the blot shown. Could the authors provide an improved exposure for this matter?

Reviewer #2:

Remarks to the Author:

The authors propose a combinatorial therapy for triple-negative breast cancers carrying p53 mutations (and/or p53-mutant cancers in general). Although interesting the results shown here do not fully support the authors' current conclusions. A series of additional experiments might help to fill this gap.

Figure 1E is currently difficult to read and interpret (i.e. there is no clear explanation of the yellow dashed-line/box present in the figure).

More importantly, the authors state that 'Expression of individual genes in BER and MMR pathways is elevated in p53-deficient tumors in predominantly TNBC/Basal-like' but do not perform/show any relevant statistical analysis.

Given the high frequency of p53 mutations in TNBC it is critical to determine whether the high expression of BER and MMR genes is an intrinsic feature of this breast cancer subtype or it is specifically associated with a p53 mutant status. The authors should compare expression levels of BER and MMR genes in p53-mutant vs p53-wt breast cancers as whole and in p53 mutant vs p53 wt cancers within the TNBC group.

The terms p53-deficient and p53-mutant are often used indistinguishably in the text. Given that missense mutations (including gain-of-function mutations) rather than loss of p53 are commonly found in TNBC, the use of these terms should be carefully revised throughout the text.

The use of MCF10A cells alone as control in the experiments shown in Figure 2, Figure 3 and Figure 4 is not sufficient to conclude that the observed phenotypes are dependent on p53 mutations. MCF10A cells carry wt p53 but are also normal-like breast cells not cancer cells. These experiments should be performed using wt p53 breast cancer cells as additional controls (i.e. MCF7 cells and/or even better triple-negative breast cancer cells with wt p53, such as MDA-MB-361 or similar).

Addition of at least another triple-negative breast cancer cells (p53-mutant or p53-deficient) in key experiments would further strengthen the authors' current conclusion.

Olaparib dose-reponse curve relative to MDA-MB-468 cells is missing in Figure 6A.

The therapeutic strategy proposed here relies on the induction of DNA damage (and possibly additional mutations) in the target tumour cells, which can in turn promote cancer progression (and might be of concern), especially if not all the cells are killed during treatment. This point should be better discussed.

No mechanistic insight is provided to explain why treatment with deoxyuridine analogues might render sensitive specifically to PARP inhibitors. Can deoxyuridine analogues induce inactivating mutations in genes involved in DNA double strand breaks?

Can treated cells efficiently repair double strand breaks? RAD51 foci staining or similar analysis might help address this question.

Interestingly, the in vivo experiment shows that tumour grafts treated with the 'combo TAS-OLA' do not grow from about day 16 to day 18 but they then stop responding and grow at a rate comparable

to control tumours or tumours subjected to single agent- treatments.

Would it be better to pre-treat tumours with TAS alone prior to a treatment with OLA alone? Or start with a combo TAS-OLA and continue with OLA alone?

Stopping the treatment with TAS early on might prevent the onset of additional mutations (including mutations able to confer resistance to OLA).

April 07, 2021

Re: COMMSBIO-20-2529-T, title: "Selective therapeutic strategy for p53-deficient cancer by targeting dysregulation in DNA repair"

Dear Reviewers,

We are grateful for the opportunity to address the Reviewer's comments. Please find below a detailed point-by-point response to all Reviewer comments, accompanied by corresponding changes to the manuscript.

In addressing the Reviewer comments, we have performed additional experiments and introduced the appropriate changes in the text. As recommended by the Reviewers, we assessed the drug responses in several p53 wild-type and mutant cells lines, including cytotoxicity data for individual drugs and their combinations, and combination index (data are presented in Figure 3C-E, Supplementary Figure 6, Supplementary Tables 1-3); evaluated DNA double-strand breaks using RAD51 foci by immunofluorescence (Figure 3G,I); provided cytotoxicity curves for PARP inhibitors in several cells (Figure 6D); measured caspase-3/7 activity in vitro and in vivo (Figure 7C and Suppl. Figure 7C-D). The manuscript now includes 7 main figures, 8 supplementary figures, and 3 supplementary tables. The supplementary material is organized in a single PDF file, entitled: "Supplementary Information". Please see the detailed response below.

Respectfully,

Andrei V. Bakin, Ph.D.
Associate Professor of Oncology
Department of Cancer Genetics and Genomics
Roswell Park Comprehensive Cancer Center
andrei.bakin@roswellpark.org

Reviewer #1

Comment 1. “The authors use a non-tumoral p53 wild-type (wt) breast epithelial cell line, MCF10A, to compare outcomes from p53-mutant breast cancer cell lines in response to the indicated treatments. This is of great value to show that the combination therapy preferentially targets p53-mutant TNBC cells over healthy cells; however, to reinforce that targeting the BER pathway is a selective therapy for p53-mutant TNBC, the work would benefit from showing at least in vitro the effect (cytotoxicity profiles, tumor growth) of treating p53-wt TNBC cells (i.e. SUM102PT cells) with PARPi and fluorinated pyrimidine analogs.”

Reply: We thank the Reviewer for this important comment. We have now performed experiments with two TNBC cell lines carrying wild-type p53: human CAL51 and mouse EMT6 (derived from spontaneous mouse mammary carcinoma, BALB/c strain). The FdUrd response data are shown in Figure 3E and 3C. For comparison of the drug response in mouse carcinoma cell lines, we also included the data for 4T1 cell line (p53-null) in Figure 3C. In addition, we included the data for lung cancer A549 cell line (p53 wild type) in Figure 3D and 5D. Thus, we have analyzed the drug response in non-tumor MCF10A cells and three p53 wild-type tumor cell lines (CAL51, EMT6, and A549). We also expanded the analysis of p53-mutant TNBC cell lines to 4T1 (Figure 3C) and BT549 (Figure 5E). The cytotoxicity data for CAL51, EMT6, and 4T1 are now included in Supplementary Tables 1-3. In addition, we have examined the EdU-removal ability in p53 wild-type CAL51 and p53 mutant MDA-MB-468 TNBC cell lines by using flow cytometry. The data are now included in Supplementary Figure 4. The data indicate that CAL51 cells efficiently remove EdU while in MDA-MB-468 the EdU removal is significantly delayed. The corresponding changes were made in the text.

Comment 2. “Accumulation of unrepaired EdUrd/FdUrd-induced DNA damage in p53-mutant breast cancer cells could lead to p53-independent cell death. The work would benefit from showing the apoptotic profiles (i.e. annexin V stain, cleaved caspase detection by western blotting) of EdUrd/FdUrd-treated MCF10A and MDA-MB-231 cells after drug wash out and during continuous treatment.”

Reply: We thank the Reviewer for this important comment. We have measured Caspase 3/7 activity in MDA-MB-231 and MCF10A cells after continuous treatment with the tested drugs for 48 hours using homogeneous luminescence assays and the data are included in Supplementary Figure 7C-D.

Comment 3. “The role of p53 in enabling repair of FdUrd-induced DNA damage is evaluated at 24 and 48h. Likewise, p21 should also be assessed at 48h for DNA damage repair in p21-wt and p21-knockout cells.”

Reply: We thank the Reviewer for this important comment. We performed additional experiments and the data on the time-course of the FdUrd response are shown in Figure 3G. The text was changed accordingly.

Comment 4. “Combination of TAS-102 and Olaparib leads to DNA damage accumulation (increased gH2AX) and reduced proliferation (decreased Ki67 levels) in p53-mutant tumors, observed by histological analyses. The work would benefit from showing also cell death (i.e. cleaved caspase 3) in the treated xenografts.”

Reply: We thank the Reviewer for this insightful comment. As recommended, we have now performed the IHC analysis of active (cleaved) Caspase-3 in xenografts, and the data are now shown in Figure 7C. The corresponding changes were made in the text.

Minor Points.

Comment 1. “Figure 6A. Cytotoxicity curve should be shown for Olaparib alone in MDA-MB-468 for comparison, since combined treatment with Olaparib and fluorinated pyrimidine analogues (FdUrd or 5-FU) is used in this cell line.

Figure 6F. Cytotoxicity curves for Talazoparib alone should be shown in MCF10A, MDA-MB-231 and MDA-MB-468 for comparison.”

Reply: We thank the Reviewer for these comments. As recommended, the cytotoxicity curves were added to Figure 6 (panels A, D).

Comment 2. “Lines 335-338. Attenuation of p53-p21 signalling and gHA2X levels in p53-wt cells in response to PARPi-FdUrd combination is not well appreciated in blots of Figure 5A, D and Supplementary Figure 3C. Please, provide a longer exposure to clearly show the differences if possible.”

Reply: We thank the Reviewer for these comments. As suggested, we now included the data that show attenuation of gHA2X levels in p53 wild-type A549 and EMT6 cell lines (Figure 5 and Supplementary Figure 6A).

Comment 3. “Figure 1A-B. For clarity, please, add the X axis line in (A) and Y axis labels (i.e. “S-phase genes expression, fold change...”; “tumor mutation burden, mutations/Mb”) in (A-B).

Supplementary Figure 1. Change (E) to (C) in the figure legend. Supplementary Figure 1C. Please, add Y axis label.”

Reply: We thank the Reviewer for these comments. We now modified the Figure 1A-C to improve clarity of the presented data.

“Supplementary Fig 1C. It is not clear to the reader if panel (C) shows gene expression levels for MDA-MB-231 or MCF10A, or if it shows the fold change of gene expression levels of MDA-MB-231 over MCF10A. I would recommend for clarity and comparison purposes to represent gene expression levels of the two cell lines in separate graphs.”

Reply: We thank the Reviewer for this comment. We now modified the Suppl. Fig. 1C to indicate the comparison of gene expression levels between MDA-MB-231 and MCF10A. The data are now presented in Supplementary Figure 2.

“Figure 2A, C. DNA content (Hoechst) should be represented in linear scale.”

Reply: The data now presented in linear scale.

“Figure 3F. Please, indicate concentration and hours of FdUrd treatment in the figure legend.”

Reply: The corresponding changes are done.

“Line 274 “EdUrd treatment induced activation and stabilization of p53 irrespective of p53 status...”. Stabilization of p53 is not observed in p53-mutant MDA-MB-231 and MDA-MB-468 cells (Figure 3A-C). Please, change sentence accordingly.”

Reply: The corresponding change was made (line 293 now): “EdUrd treatment induced phosphorylation of p53 in both cell lines...”

“Figure 4D. For clarity, please, indicate Olaparib and Talazoparib concentrations, instead of a concentration range, in panel (D) and in the figure legend.”

Reply: The corresponding change was made in the legends.

“Figures 3F and 5F. Add scale bars to microscopy images.”

Reply: The scale bars are included in the Figures.

“Figure 5F. Indicate hours of treatment in the figure legend.”

Reply: The time is added.

“Figure 6A-D, F. Axis labels and legend in panels are non-readable, please, replaced this figure by a version with better resolution.”

Reply: We apologize for this. The corresponding changes were made.

“Figure 6E. Add axis labels and legend in panel (E).”

Reply: The axis labels are added.

“Figure 6. Remove panel (G) mention in the figure legend. There is no panel (G) in the actual figure or in the text.”

Reply: We made the corrections. In addition, as recommended by the Reviewer 1, we included the cytotoxicity data for Talazoparib (Figure 6D), and panel 6F is now panel 6G.

“Supplementary Figure 3. Panel (C) is referred in the text (line 338) earlier than panel (B) (line 381). Please, format supplementary figure for consistency with the text.”

Reply: The corresponding changes were made in the text and figure arrangement.

Comment 4. “Figure 3E. p21wt MCF10A cells are treated with FdUrd 5 uM for 24h, as in Figure 3B and 3D (siRNA-SCR); however, upregulation of gH2AX levels at this timepoint is not observed, in contrast to what previously shown. Please, show a more representative blot of this experiment in line with previous observations or explain in the text why gH2AX levels are not increased in this case.”

Reply: We thank the Reviewer for this important comment. We now show more representative blot images in panels 3B, 3F, and 3G for MCF10A cells, including p21wt MCF10A (panel 3G).

Comment 5. “Figures 3F. Could the authors explain the discrepancy in nuclear size of CTRL vs FdUrd-treated MDA-MB-231 cells? Please, show a more representative image of CTRL MDA-MB-231 cells.”

Reply: We thank the Reviewer for this comment. As recommended by the Reviewer, we provide more representative images of cells treated with FdUrd or Doxorubicin, and stained for RAD51 (red), γ H2AX (green) and DNA (Hoechst, blue). Images and scale bars are now presented in Figure 3G.

Comment 6. “Levels of Olaparib-induced p21 activation are inconsistent across experiments at the same timepoint (24h) and concentration of drug used (0.5 uM; Figure 4B and 5A); p21 protein upregulation is minimal in MCF10A cells from Figure 5A. To help follow the rationale of the work better, I would suggest to provide blots that reflects more accurately and consistently Olaparib-induced p21 activation

across experiments.”

Reply: We thank the Reviewer for this comment. Now Figure 4B and Figure 5A show the immunoblots for p21 in MCF10A that more accurately and consistently reflect Olaparib-induced p21 activation. In addition, we show in Figure 5D that Olaparib increases p21 levels in A549 cells (p53 wild-type).

Comment 7. “Previous works (i.e. PMID: 25700705) have shown that TAS-102 induces p53-p21 signalling with minimal double strand breaks, please, contextualize findings in line with literature.”

Reply: We thank the Reviewer for this insightful comment. We now put our findings into the context with the prior work. The corresponding changes were made in the text (lines 509-511).

Comment 8. “Supplementary Figure 3A. p53 stabilization and p21 upregulation in Olaparib-treated WI-38 (p53 wt) cells is not very clear in the blot shown. Could the authors provide an improved exposure for this matter?”

Reply: We now provide the data for PARPi-induced activation of p53-p21 signaling in two p53 wild-type cell lines (A549 and CAL51), in addition to WI-38 cells. The data are shown in Suppl. Figure 6 for Olaparib and Talazoparib.

Reviewer #2

“The authors propose a combinatorial therapy for triple-negative breast cancers carrying p53 mutations (and/or p53-mutant cancers in general). Although interesting the results shown here do not fully support the authors' current conclusions. A series of additional experiments might help to fill this gap.”

Comment 1. “Figure 1E is currently difficult to read and interpret (i.e. there is no clear explanation of the yellow dashed-line/box present in the figure). More importantly, the authors state that ‘Expression of individual genes in BER and MMR pathways is elevated in p53-deficient tumors in predominantly TNBC/Basal-like’ but do not perform/show any relevant statistical analysis.

Given the high frequency of p53 mutations in TNBC it is critical to determine whether the high expression of BER and MMR genes is an intrinsic feature of this breast cancer subtype or it is specifically associated with a p53 mutant status. The authors should compare expression levels of BER and MMR genes in p53-mutant vs p53-wt breast cancers as whole and in p53 mutant vs p53 wt cancers within the TNBC group.”

Reply: We thank the Reviewer for this important comment. As recommended by the Reviewer, we now included the statistical data for BER and MMR gene expression in respect to the p53 status in Breast Cancer TCGA dataset (Suppl. Figure 1C). We have also clarified that the yellow box indicates the expression cluster (with reference to the dendrogram to the left) of BER/MMR expression, p53 mutants and basal-like cases.

We did not observe a statistically significant correlation when the BER/MMR/RRG gene-sets were compared with p53 status within the TNBC/Basal-like group alone (Suppl. Figure 1D). It is possible that the high expression level (a maximum achievable level) of the tested genes and/or a small group size of the p53 wild-type subset in the TNBC group contribute to the lack of the p53 status discrimination. While this finding may suggest that expression of BER/MMR genes is an intrinsic feature of TNBC/Basal-like group, the data also indicate that BER/MMR gene expression closely correlates with expression of replication-related genes (RRGs) which are primarily elevated in the TNBC/Basal-like

group, Figure 1D, E. Thus, a correlation in the expression of BER/MMR and RRG genes may represent a feature of p53 mutant cancers, including TNBC.

Comment 2. “The terms p53-deficient and p53-mutant are often used indistinguishably in the text. Given that missense mutations (including gain-of-function mutations) rather than loss of p53 are commonly found in TNBC, the use of these terms should be carefully revised throughout the text.”

Reply: We thank the Reviewer for this insightful comment. We used term “deficiency” when p53 gene is deleted or not expressed (like 4T1 cell line or HCT116 line with p53 knockout). For consistency, we now use “mutant” throughout the text.

Comment 3. “The use of MCF10A cells alone as control in the experiments shown in Figure 2, Figure 3 and Figure 4 is not sufficient to conclude that the observed phenotypes are dependent on p53 mutations. MCF10A cells carry wt p53 but are also normal-like breast cells not cancer cells. These experiments should be performed using wt p53 breast cancer cells as additional controls (i.e. MCF7 cells and/or even better triple-negative breast cancer cells with wt p53, such as MDA-MB-361 or similar). Addition of at least another triple-negative breast cancer cells (p53-mutant or p53-deficient) in key experiments would further strengthen the authors’ current conclusion.”

Reply: We thank the Reviewer for this insightful comment. As recommended by the Reviewer, we now included the data for several cell lines with wild-type p53 in addition to non-tumor human breast epithelial cell line MCF10A, i.e. human TNBC cell line CAL51, human lung carcinoma cell line A549, and mouse mammary carcinoma cell line EMT6 (Figures 3, 5, Supplementary Figures 4, 6, 7). All these cell lines showed transient activation of DNA damage signaling (phospho-Ser139-H2AX), in contrast to p53-mutant, p53-null, or p53-depleted cell lines.

Comment 4. “Olaparib dose-response curve relative to MDA-MB-468 cells is missing in Figure 6A.”

Reply: We thank the Reviewer for this comment. The data now included in Figure 6A. In addition, we also included cytotoxicity curves for Talazoparib in Figure 6D.

Comment 5. “The therapeutic strategy proposed here relies on the induction of DNA damage (and possibly additional mutations) in the target tumour cells, which can in turn promote cancer progression (and might be of concern), especially if not all the cells are killed during treatment. This point should be better discussed.”

Reply: We thank the Reviewer for this comment. The mechanism is now discussed in the last two paragraphs of the text. The proposed therapeutic strategy is based on the identified defect in BER-mediated removal of deoxyuridine analogues incorporated into DNA at the A-U pair. In the first step of BER, DNA glycosylases (UNG, TDG) remove the incorporated modified analogues, and this step effectively works in both p53 wild-type and p53 mutant cells (based on the data in Figure 2 and Supplementary Figure 4). In p53 mutant cells, DNA damage fails to activate the p53-p21 axis and DNA replication arrest, leading to formation of double-strand breaks (Figures 3, 5). PARP inhibitors induce G1 arrest in p53 wild-type cells and G2 accumulation in p53 mutant cells, apparently due to a delay in repair of DNA single-strand breaks generated in the first step of BER. Interestingly, knockout of UNG does not increase mutation rates, but leads to accumulation U-residues in DNA. Thus, incorporation of uracil residue may not increase mutational rate. Nonetheless, we agree with the Reviewer that DNA damage induced by BER may lead to mutations in p53 mutant cancer cells. This is actually a common issue of nearly all anti-cancer treatment strategies, including radiation therapy, chemotherapy (cisplatin,

doxorubicin, taxanes) and targeted therapy such as directed against cell cycle regulators (CDK4/6 inhibitors). In fact, many of the above-mentioned treatments lead to mutations underlying therapy resistance and tumor recurrence. Thus, this important aspect of treatment merits further investigation.

Comment 6. “No mechanistic insight is provided to explain why treatment with deoxyuridine analogues might render sensitive specifically to PARP inhibitors. Can deoxyuridine analogues induce inactivating mutations in genes involved in DNA double strand breaks? Can treated cells efficiently repair double strand breaks? RAD51 foci staining or similar analysis might help address this question.”

Reply: We thank the Reviewer for this insightful comment. As recommended by the Reviewer, we have performed RAD51 foci evaluation in p53 wild-type and mutant cells. These experiments confirmed our initial finding with gH2AX foci staining, namely, p53 wild-type cells are capable of DNA double strand break repair in response to deoxyuridine analogue, whereas DNA damage is accumulated in p53 mutant cancer cells. The data now included in Figure 3G-I.

In respect to the mechanism, our study indicates that PARP inhibitors enhance anti-tumor activity of deoxyuridine analogues. PARP inhibition delays repair of DNA breaks introduced into DNA after removal of deoxyuridine analogue in the first step of BER. In p53 mutant cells, DNA damage fails to activate the p53-p21 axis and DNA replication arrest, leading to formation of double-strand breaks. Further, the caspase-3/7 activity data showed that the drug combination enhances caspase activity compared to control and each drug alone (Suppl. Figure 7C-D). Note, PARP inhibitors are given at the concentrations that block PARP activity but not toxic for non-tumor and cancer cells. PARP inhibitors induced G1 arrest in normal p53 wild-type cells while causing G2 accumulation in p53-mutant cells. Hence, the combination of two drugs leads to G1 arrest in p53 wild-type cells, while enhances DNA damage and death in p53 mutant cells. This model is consistent with the DNA damage data (Figures 3 and 5) cytotoxicity data (Figure 6), caspase-3/7 activity data (Suppl. Figure 7C-D), and xenograft study data (Figure 7). The model is discussed in the last two paragraphs of the text. We agree that DNA damage induced by the regimen may, potentially, provide a route for therapy resistance and tumor recurrence, a common problem of nearly all anticancer therapy, this aspect merits further investigation.

Comment 7. “Interestingly, the in vivo experiment shows that tumour grafts treated with the ‘combo TAS-OLA’ do not grow from about day 16 to day 18 but they then stop responding and grow at a rate comparable to control tumours or tumours subjected to single agent- treatments.

Would it be better to pre-treat tumours with TAS alone prior to a treatment with OLA alone? Or start with a combo TAS-OLA and continue with OLA alone?

Stopping the treatment with TAS early on might prevent the onset of additional mutations (including mutations able to confer resistance to OLA).”

Reply: We thank the Reviewer for this insightful comment. Our treatment strategy is based on the study data, i.e. the drug combination showed higher selective toxicity towards p53-mutant cells in respect to DNA damage and higher caspase-3/7 activity. The alternative strategy is pre-treatment with PARPi and then followed by treatment with TAS102, since PARP inhibitor promotes G1 arrest in p53wt cells and does not reduce incorporation of deoxyuridine analogues (FdUrd, TFT) into DNA in p53 mutant cells. In this way, TAS102 incorporation will be selective to cancer, while normal tissue will be protected. If TAS102 is given at first, then the protective effect of PARPi in normal tissues will be limited. The in vivo data also suggest a treatment strategy with limited exposure to TAS102, i.e. treatment with the drug combination initially, and then continue with PARPi alone. Thus, further studies are needed to evaluate those and other strategies, based on the drug combination synergism identified in the current work.

Reviewers' Comments:

Reviewer #1:

None

Reviewer #2:

Remarks to the Author:

The authors performed a series of additional experiments in order to address all the points raised during the first round of revision.

Inclusion of additional details regarding Western blot analysis and relative quantification would be beneficial.

Elm & Carlton Streets | Buffalo, New York 14263
1-800-ROSWELL (1-800-767-9355)
RoswellPark.org | AskRoswell@RoswellPark.org

May 27, 2021

Re: COMMSBIO-20-2529B, title: "Selective therapeutic strategy for p53-deficient cancer by targeting dysregulation in DNA repair"

Dear Reviewers,

We are grateful for the opportunity to address the Reviewer's comments. In addressing the Reviewer comments, we have added a description of Immunoblotting to the Methods section. Please see the detailed response below.

Respectfully,

Andrei V. Bakin, Ph.D.
Associate Professor of Oncology
Department of Cancer Genetics and Genomics
Roswell Park Comprehensive Cancer Center
andrei.bakin@roswellpark.org

Reviewer #2

“The authors performed a series of additional experiments in order to address all the points raised during the first round of revision.”

Comment 1. “Inclusion of additional details regarding Western blot analysis and relative quantification would be beneficial.”

Reply: We thank the Reviewer for this important comment. We apologize for not including this information. As recommended by the Reviewer, we now included a description of Immunoblotting to the Methods section.